# α-Synuclein emulsifies TDP-43 prion-like domain—RNA liquid droplets to promote heterotypic amyloid fibrils

Shailendra Dhakal[1,2], Malay Mondal[1,2], Azin Mirzazadeh[1,2], Siddhartha Banerjee[3], Ayanjeet Ghosh [3] & Vijayaraghavan Rangachari [1,2✉]

Many neurodegenerative diseases including frontotemporal lobar degeneration (FTLD), Lewy body disease (LBD), multiple system atrophy (MSA), etc., show colocalized deposits of TDP-43 and α-synuclein (αS) aggregates. To understand whether these colocalizations are driven by specific molecular interactions between the two proteins, we previously showed that the prion-like C-terminal domain of TDP-43 (TDP-43PrLD) and αS synergistically interact to form neurotoxic heterotypic amyloids in homogeneous buffer conditions. However, it remains unclear if αS can modulate TDP-43 present within liquid droplets and biomolecular condensates called stress granules (SGs). Here, using cell culture and in vitro TDP-43PrLD – RNA liquid droplets as models along with microscopy, nanoscale AFM-IR spectroscopy, and biophysical analyses, we uncover the interactions of αS with phase-separated droplets. We learn that αS acts as a Pickering agent by forming clusters on the surface of TDP-43PrLD – RNA droplets. The aggregates of αS on these clusters emulsify the droplets by nucleating the formation of heterotypic TDP-43PrLD amyloid fibrils, structures of which are distinct from those derived from homogenous solutions. Together, these results reveal an intriguing property of αS to act as a Pickering agent while interacting with SGs and unmask the hitherto unknown role of αS in modulating TDP-43 proteinopathies.

[1] Department of Chemistry and Biochemistry, School of Mathematics and Natural Sciences, University of Southern Mississippi, Hattiesburg, MS 39406, USA. [2] Center for Molecular and Cellular Biosciences, University of Southern Mississippi, Hattiesburg, MS 39406, USA. [3] Department of Chemistry and Biochemistry, University of Alabama, Tuscaloosa, AL 35401, USA. ✉email: vijay.rangachari@usm.edu

Tar-DNA binding protein 43 (TDP-43) and α-synuclein (αS) are two proteins involved in the formation of amyloid deposits observed among several neurodegenerative diseases known as TDP-43 proteinopathies and α-synucleinopathies, respectively[1,2]. TDP-43 proteinopathies include pathologies such as frontotemporal lobar degeneration (FTLD), and amyotrophic lateral sclerosis (ALS), in which the cytoplasmic amyloid inclusions of TDP-43 and its c-terminal fragments are known to contribute to the etiology[1,3]. α-synucleinopathies, which include Parkinson's disease (PD), multiple system atrophy (MSA), and dementia with Lewy bodies (DLB)[4], involve deposits in the form of Lewy bodies comprised of αS aggregates that are predominantly present in the cytosol. Lately, an increasing number of pathologies including limbic age-related TDP-43 encephalopathies (LATE), FTLD, and MSA exhibit clinical, pathological, and genetic overlaps and often show colocalized TDP-43 and αS deposits, which are thought to contribute to the phenotypes and clinical presentations observed in these pathologies[5,6]. The colocalization of the two proteins observed in the aforementioned pathologies implores the question of whether the two proteins engage in specific molecular-level interactions.

TDP-43 is a ribonucleoprotein involved in a variety of functions such as transcriptional regulation, RNA biogenesis, and stress response[7,8]. In pathologies, TDP-43 translocates into the cytoplasm and undergoes aberrant cleavage to form c-terminal fragments ranging from 35 to 16 kDa fragments that form neurotoxic amyloid inclusions[9–12]. On the other hand, during the cell's response to stress, TDP-43 translocates to the cytosol by sequestering mRNA along with other proteins to partition into membraneless organelles called stress granules (SGs), thereby controlling gene regulation. Although SGs are reversibly formed protein-RNA condensates, during chronic stress conditions, it is believed that SGs could become a nidus for amyloid formation[13–15]. Therefore, understanding the molecular factors and mechanisms that influence SGs could provide clues in the onset and progression of neurodegenerative diseases. Based on proximity, one can conjecture that the presence of αS in the cytoplasm could potentiate the modulation of TDP-43 aggregation pathways, and the underlying molecular interactions could explain the colocalization of the two proteins and the phenotypes observed in the aforementioned pathologies. This hypothesis was supported by our previous work that showed the ~16 kDa prion-like c-terminal domain of TDP-43 (TDP-43PrLD) interacts with αS to form distinctive heterotypic hybrid fibrils containing stoichiometric proportions of two proteins in homogenous buffers[16]. Interestingly, polymorphic heterotypic fibrils emerge depending on the nature of aggregate seeds used for cross-interactions, i.e., seeding of TDP-43PrLD monomers with either αS monomers or fibrils generates structurally distinguishable TDP-43PrLD fibrils[17]. Our preliminary investigations also provided clues on the possible modulation of SGs by αS using TDP-43PrLD-RNA droplets as models, but details of the mechanism remain unknown[16].

Here, we take a deep dive to uncover the interactions between TDP-43PrLD and αS both using cell culture models and in vitro TDP-43PrLD-RNA liquid droplets that led us to make surprising and noteworthy observations on the interaction dynamics of αS with TDP-43PrLD-RNA droplets. Using confocal microscopy, AFM-IR spectroscopy, and other biophysical techniques, we observe that only a part of bulk αS partitions into TDP-43PrLD-RNA droplets, largely remaining as clusters on the surface of the droplets acting as Pickering agents. The subsequent engulfment of the TDP-43PrLD—RNA droplets by αS leads to emulsification by nucleating heterotypic amyloid formation. Our in vitro results establish a possible molecular mechanism involved in the colocalization of αS with TDP-43PrLD puncta both under stress and

non-stress conditions in cells, potentiating the formation of insoluble cytoplasmic heterotypic amyloid inclusions. These results bring forth detailed molecular events involving the interaction of αS with TDP-43, present within the coacervated droplets with RNA. The concomitant promotion of heterotypic amyloids with distinct structures answers some key questions on how αS may influence SGs to make them hubs for toxic heterotypic amyloids formation and TDP-43 proteinopathies at large.

## Results

### αS colocalizes with cytoplasmic TDP-43PrLD puncta under both stressed and non-stressed conditions in HeLa cells.

Aberrantly cleaved C-terminal fragments of TDP-43 are known to form toxic inclusions in the cytoplasm[9–12]. These fragments range from 16 to 35 kDa in which the 16 kDa TDP-43PrLD constitutes a major part that drives phase separation or aggregation[18,19]. To see the potential colocalization of αS with TDP-43PrLD, the two proteins were co-transfected in HeLa cells. Transiently expressed proteins were then investigated in both live and fixed cells. Live cell confocal images both in stressed (Fig. 1a) and non-stressed (Fig. 1c) cells showed co-localization of TDP-43PrLD and αS in the cytosol (arrows). Fluorescence resonance recovery after photobleaching (FRAP) analysis on the colocalized puncta presumed to be SGs, showed partial recovery (~60%) with TDP-43PrLD, suggesting diminished internal mobility or in other words, gelation (red, ■; Fig. 1b). In contrast, αS showed no recovery in these puncta which suggests that it could be in an aggregated state (green, ●; Fig. 1b). Under non-stressed conditions, FRAP on the colocalized puncta showed significant attenuation of recovery for TDP-43PrLD (~40%) (red, ■; Fig. 1d) while αS did not show any (green, ●; Fig. 1d). These results suggest that while under non-stressed conditions when no SGs are present, αS and TDP-43PrLD co-localize as puncta in the cytosol possibly as heterotypic aggregates as demonstrated earlier[17], under stress conditions, αS colocalizes with TDP-43PrLD present in SGs. The cells were also fixed and investigated by immunochemistry with TDP-43, αS, and TIA-1 antibodies. TIA-1, known to partition into SGs was used as a biomarker (Fig. 1e–g). Under stress conditions, two types of colocalizations were observed; a two-way colocalization between TDP-43PrLD and TIA-1 that is consistent with partitioning into SGs (Fig. 1e), and a three-way colocalization between TDP-43, αS, and TIA-1 (Fig. 1f). As expected, in non-stress conditions, the two proteins showed colocalized in the cytosol and not in SGs (TIA-1-negative) correlating with live cell imaging (Fig. 1g). While the two-way colocalization of TDP-43PrLD and TIA-1 was predominant with over 54% of the co-expressed cells, three-way colocalization was observed among 6% of the cells under stressed conditions. Under stress conditions, the colocalization of TDP-43 and TIA-1 was greater (red; Fig. 1h). Similarly, colocalization between TDP-43PrLD and αS was greater in stress than in non-stress conditions (green; Fig. 1h), suggesting αS interacts with TDP-43PrLD both in SGs and cytosol.

### A fraction of αS in the solution slowly partitions onto the surface of TDP-43PrLD – RNA liquid droplets in vitro.

Encouraged by the data obtained from cell culture experiments on the colocalization of αS and TDP-43PrLD, to determine if αS partitions into TDP-43PrLD – RNA droplets, we initially assessed the propensity of the two proteins to phase separate. In silico analysis by FuzDrop[20–22] showed TDP-43PrLD has a high droplet-promoting probability ($P_{DP}$ score) of near unity across most of its primary sequence (Fig. 2a). αS, on the other hand, showed high $P_{DP}$ scores only in certain sections of the protein with the c-terminal domain (CTD) showing higher propensity than the N-terminal domain (NTD) while the central NAC

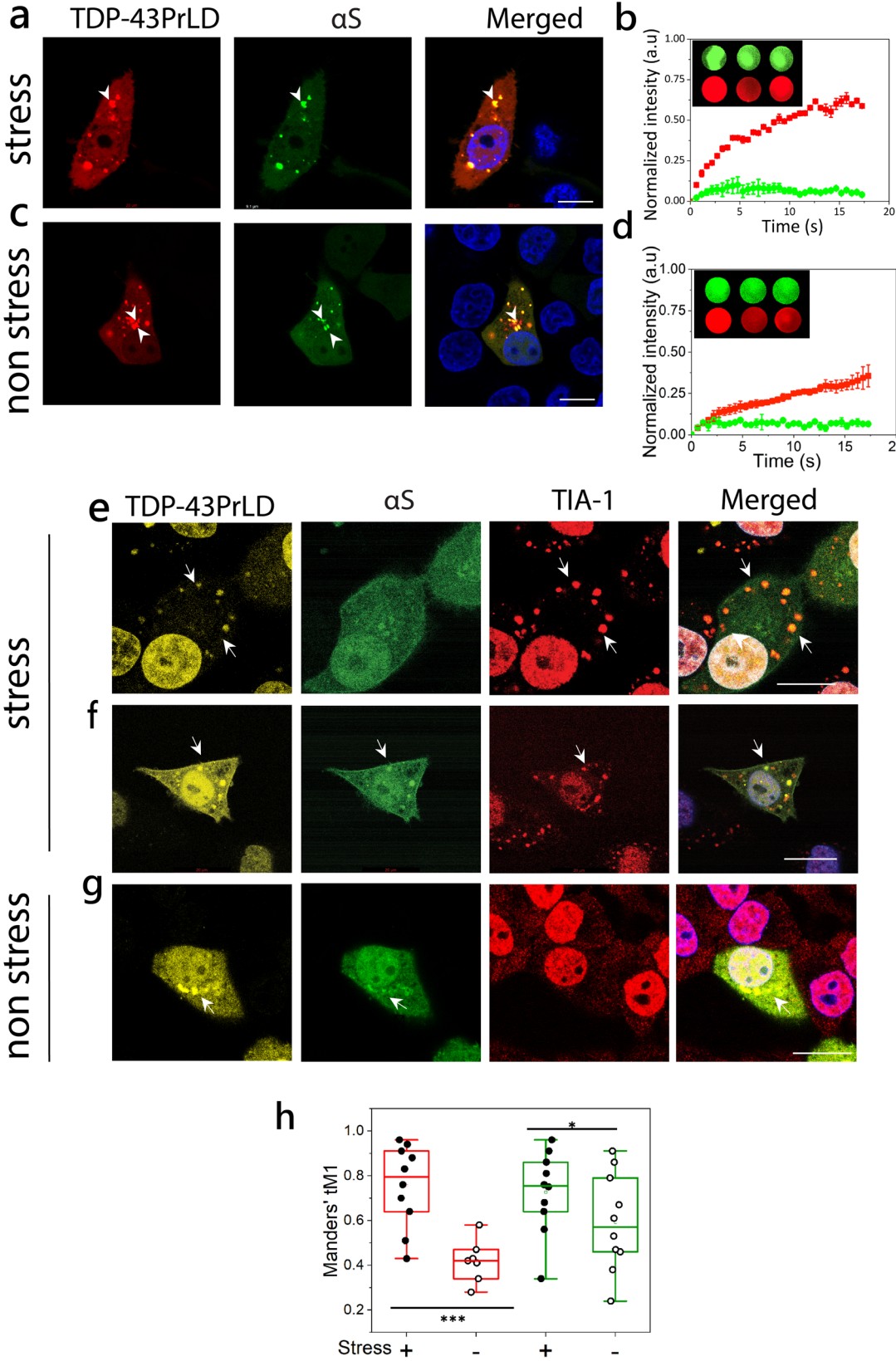

domain showed the least propensity (Fig. 2b). These predictions were confirmed by experiments also; TDP-43PrLD showed robust phase separation upon complex coacervation with poly-A RNA counter ions (Fig. 2c). Liquid-like characteristics of these droplets were further confirmed by FRAP analysis which showed a complete recovery within eight seconds (Fig. 2d). αS on the other

hand showed no droplet formation with poly-A RNA (Fig. 2e). To assess the interaction between αS and TDP-43PrLD-RNA droplets, freshly purified and fluorescently labeled αS monomers were dispensed onto the heterogenous buffer solution containing pre-formed TDP-43PrLD-RNA droplets. The time-lapse confocal microscopy images of TDP-43PrLD-RNA droplets in the

**Fig. 1 Cytoplasmic colocalization of TDP-43PrLD and αS under stress and non-stress conditions. a–d** Confocal images of live HeLa cells co-expressing tdTomato-tagged TDP-43PrLD and GFP-tagged αS. **a** Cytoplasmic colocalization of TDP-43-PrLD and αS under treatment with sodium arsenite (0.5 mM, 20 min) to induce stress. **c** And under normal conditions (non-stress), arrows indicate a few of the colocalized puncta before and after arsenite treatment. Scale bar: 20 μm. **b–d** Normalized intensities of fluorescence recovery after photobleaching (FRAP) of colocalized TDP-43PrLD (red) and αS (green) under stress (**b**) and non-stress conditions (**d**). The recovery curves of at least three independent experiments were averaged and normalized. The inset shows representative pre-bleach, post-bleach, and recovery of the region of interest (ROI). **e–g** Immunostaining of the co-transfected TDP-43PrLD and αS HeLa cells along with the stress granule (SG) marker, TIA-1 under stress and non-stress conditions. **e** TDP-43PrLD colocalizes with TIA-1 in the cytoplasm (arrows) consistent with their presence in SGs. αS did not show localization in SGs in these cells. **f** Colocalization of TDP-43PrLD, αS, and TIA-1 in the stressed Hela cells (arrows) Scale bar: 20 μm. **g** Colocalization of TDP-43PrLD and αS in the cytoplasm in non-stressed conditions. A total of 33 positive cells for co-expression were analyzed ($n = 33$). **h** Whisker plot illustrating Manders' tM1 coefficient for colocalized TDP-43PrLD, and αS. The red boxes denote immunostaining of colocalized TDP-43PrLD and TIA-1 during stress (solid circles) and non-stress (open circles). The green boxes denote the colocalization of TDP-43PrLD and αS under stress (solid circles) and under non-stress (open circles) conditions. A two-way ANOVA with $p* < 0.001$ and $p*** < 0.01$. All images were processed using FIJI (image J) and puncta were counted using ComDet plugin, and Gaussian blur filter was utilized to enhance puncta visibility. JACoP plugin used for Manders' tM1 and tM2 coefficient calculation for each channel, each data point represents the value of an independent cell ($n = 10$ and 7(for non-stress live cells)).

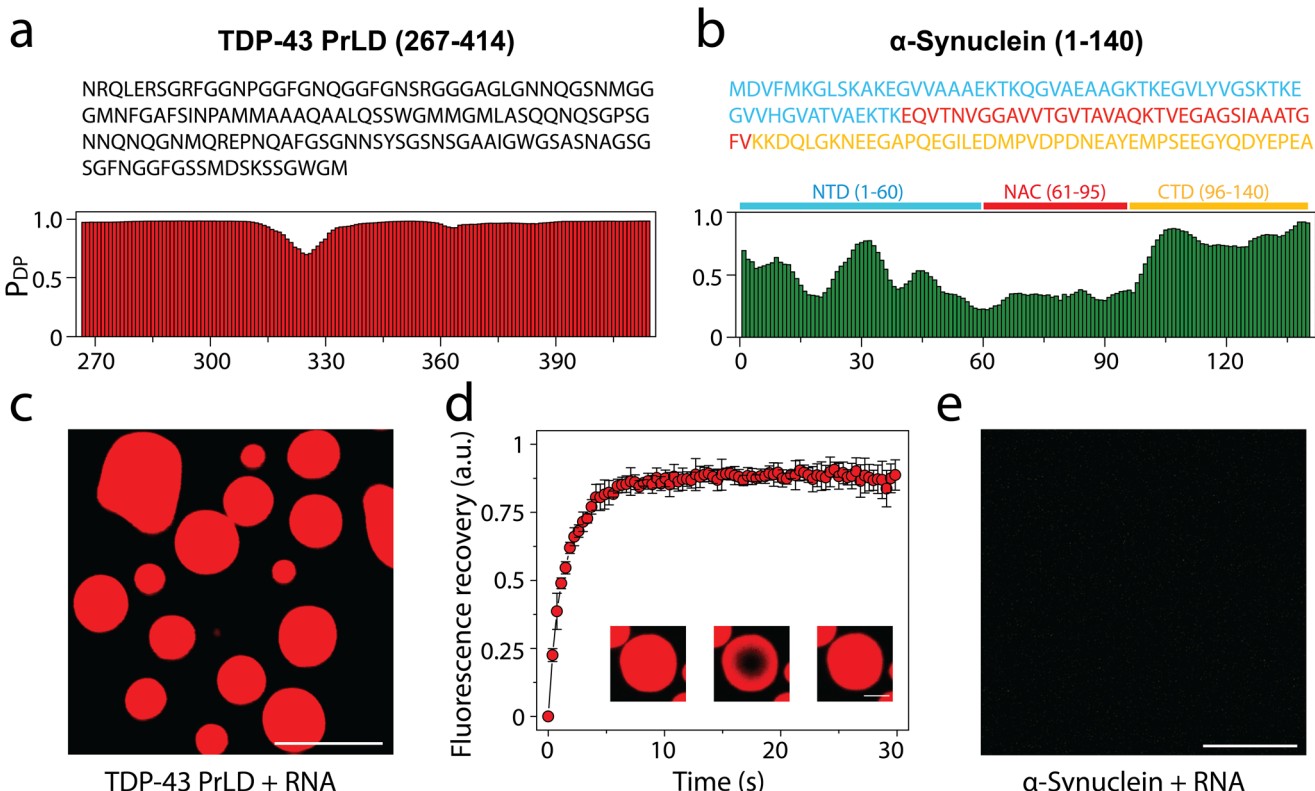

**Fig. 2 Droplet-promoting propensities of TDP-43PrLD and αS and their respective phase separation with RNA. a** Sequence of TDP-43PrLD (267-414) along with residue-based droplet-promoting probabilities ($P_{DP}$) calculated using *FuzDrop*. **b** Sequence of full-length αS showing three distinct regions; N-terminal domain (NTD; blue), non-amyloid component (NAC; red), and C-terminal domain (CTD; yellow) corresponding to 1-60, 61-95, and 96-140 residues, respectively along with residue-based droplet-promoting probabilities ($P_{DP}$). Confocal microscopic image showing phase separation of 15 μM fluorescently labeled TDP-43PrLD in the presence of poly-A RNA (**c**), and FRAP of the droplets. Insets (left to right) show pre-bleached, bleached, and post-bleached liquid droplets, respectively (**d**). **e** Confocal image of 15 μM αS in the presence of RNA that did not undergo phase separation. All the experiments were carried out in 20 mM MES buffer, pH 6.0 ($n = 3$). The scale bars in (**c**) and (**e**) are 5 μm, and the one in (**d**) is 2 μm.

presence and absence of αS show stark differences; while TDP-43PrLD—RNA droplets in the absence of αS showed coalescence within nine minutes of incubation, indicating liquid-like characteristics (Fig. 3a; arrows), those in the presence of αS indicated no coalescence (Fig. 3b; arrows). Moreover, αS seems to confine along the periphery of TDP-43PrLD – RNA droplets (Fig. 3b; arrows). Histidine-tagged proteins were used in all our studies as prohibitively low yields after the tag removal made it difficult for investigations. However, the tag did not influence the observations as the tag-free proteins showed an identical behavior (Fig. S1). To further understand the phase behavior of the three-

component system containing TDP-43PrLD—RNA—αS droplets, we sought to determine the phase boundary conditions by turbidity analyses on TDP-43PrLD with increasing RNA and αS concentrations. As expected, the turbidity values increase with the increase in RNA concentrations and peak at 75 μg/mL in the absence of αS (grey, ●; Fig. 3c). Changes in αS concentrations did not alter this pattern to any significant extent but marginally increased the turbidity values between 50 and 100 μg/mL RNA (blue, green & red, Fig. 3c). Next, by determining the turbidity of TDP-43PrLD—RNA—αS as a function of temperature, we established the upper critical solution temperature (UCST) for

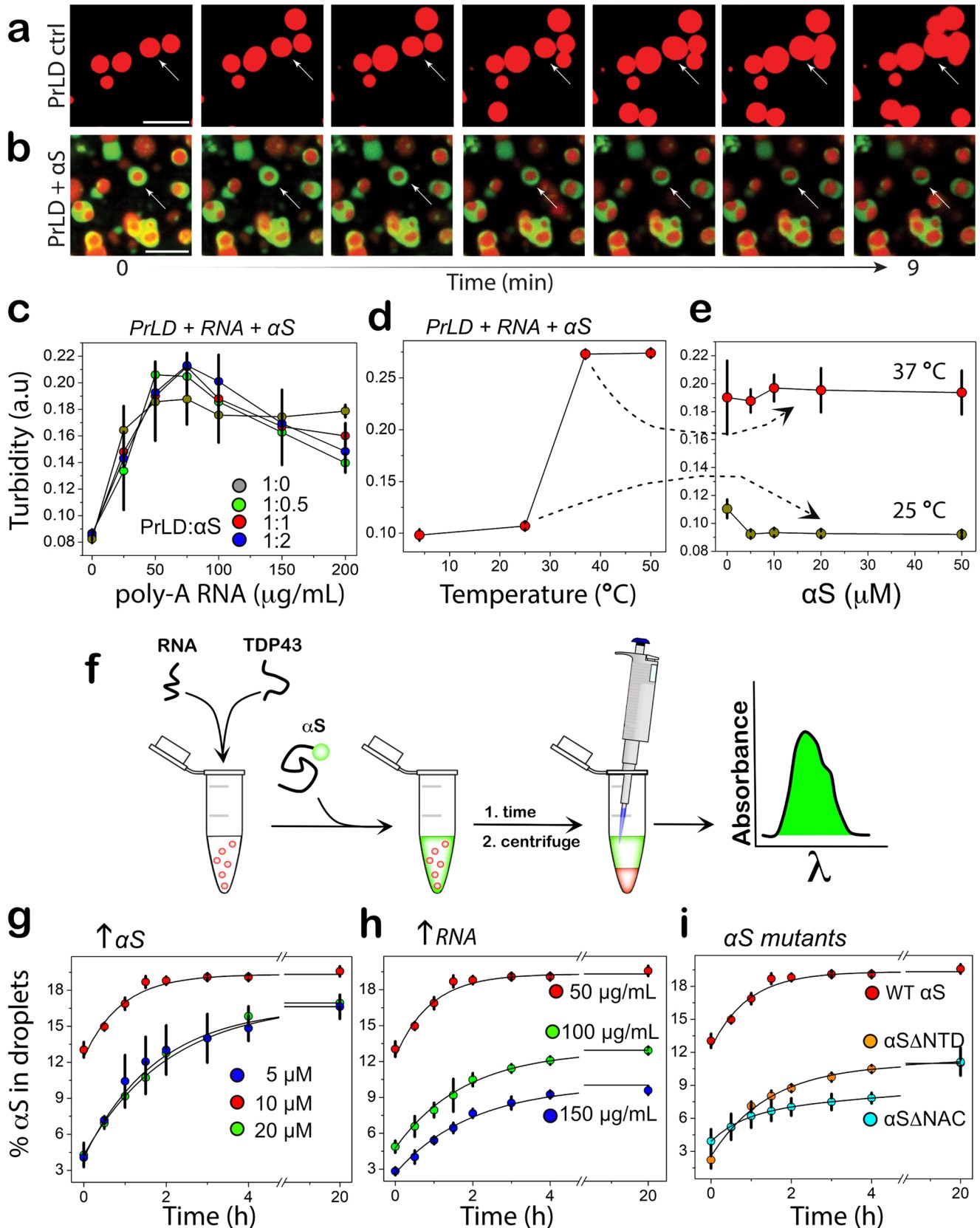

phase separation to be close to 37 °C (Fig. 3d). Furthermore, concentration dependence of the phase transitions above and below UCST (25 and 37 °C) showed no changes with αS concentration between 10 and 50 μM (Fig. 3e). To further understand the mechanism of αS partitioning, we developed a simple quantitative kinetic assay that involves measuring the amount of αS in the bulk solution and thereby calculating the precise amount of αS partitioned within the droplets (Fig. 3f). The data thus obtained for wild-type αS showed relatively slow kinetics with maximal partitioning occurring within two hours of

**Fig. 3 Phase boundary and kinetics of αS partitioning into TDP-43PrLD-RNA droplets. a** Time-lapse images of TDP-43-RNA droplets showing coalescence (arrow), characteristic of liquid-like behavior. **b** Time-lapse images of TDP-43-RNA droplets (red) in the presence of αS (green) showing coarsening with inhibition of coalescence (arrows). Yellow areas indicate merge between TDP-43PrLD and αS. The scale bars in (**a**) and (**b**) are 5 μm. **c** Turbidity measurements based on absorbance measured at 600 nm on 10 μM TDP-43PrLD samples with increasing poly-A RNA concentrations (0–200 μg/mL) and increasing αS concentrations (grey, 0 μM; green, 5 μM; red, 10 μM, and blue, 20 μM), (n = 3). **d** The turbidity measurements at different temperatures (0–50 °C) for equimolar incubation of 10 μM TDP-43PrLD and αS and with 50 μg/mL poly-A RNA (n = 3). **e** αS concentration dependence on the turbidity of 10 μM TDP-43PrLD with 50 μg/mL poly-A RNA at 25 and 37 °C. **f** Schematic of the quantitative kinetics assay performed to obtain results presented in (**g**–**i**) detailed in the Methods section. Percentage of αS partitioning into droplets as a function of time and in increasing αS concentration (5 to 20 μM) (**g**), increasing poly-A RNA concentrations (50–150 μg/mL) (**h**), and for αSΔNTD and αSΔNAC truncation mutants (**i**). All the reactions were carried out in 20 mM MES buffer at pH 6.0, and the kinetics data were fit with first-order exponential growth (n = 3).

incubation (Fig. 3g). To see whether the partitioning is dependent on αS concentration and/or stoichiometry, we varied these two parameters. Equimolar incubations with 10 μM αS and TDP-43PrLD showed only 19% of total αS partitioning in the droplets (red; Fig. 3g), while sub-stoichiometric (5 μM) and excess (20 μM) αS showed 17% (blue and green; Fig. 3g) and were statistically insignificant from one another. The partitioning of αS with insignificant quantitative change despite stoichiometric variance could be attributed to the protein's preferential localization on the surface of preformed TDP-43PrLD droplets preventing further recruitment from the bulk/dilute phase. Next, the degree of partitioning was determined as a function of RNA concentrations which showed a negative correlation between the two; 50 μg/mL of RNA (red) showed maximal partitioning with 19% while incubations with 100 and 150 μg/mL of RNA showed 12 and 8%, respectively (green and blue; Fig. 3h), suggesting charge-repulsion as a possible contributor for partitioning. To see what sequence determinants in αS are responsible for partitioning into TDP-43PrLD – RNA droplets, truncation constructs αSΔNTD and αSΔNAC were used. The N-terminal domain (NTD; 1-60) is highly charged and the central amyloid region (NAC; 61-95) is rich in hydrophobic residues therefore, deletion of these regions could provide clues about the contributions of electrostatic and hydrophobic forces, respectively. Both truncation mutants decreased the amount of αS partitioning by half (~ 9%) compared to the wild-type (red; Fig. 3i). Kinetics of partitioning, however, showed differences with αS ΔNTD displaying first-order exponential rate and αS ΔNAC showing second-order fits containing rapid and slower kinetic events with an overall slower rate than αS ΔNTD (cyan and orange; Fig. 3i). Unfortunately, we could not collect data with the acidic C-terminal truncation of αS as this construct posed problems in expression due to significant proteolytic degradation as reported previously[23]. These results suggest that both hydrophobic and electrostatic interactions differentially contribute to the partitioning of αS to the TDP-43PrLD – RNA droplets.

**αS functions as a Pickering agent in preventing TDP-43PrLD —RNA droplet coalescence.** The surprising observation that only 19% of the total αS partitions into TDP-43PrLD—RNA droplets prompted us to investigate the partitioning behavior by confocal microscopy. The control TDP-43PrLD—RNA droplets in the absence of αS showed the formation of droplets immediately as expected, which coalesced extensively with one another in 24 hours while maintaining their liquid-like behavior during this time as assessed by FRAP (Fig. 4a). Immediately upon the addition of αS to TDP-43PrLD—RNA droplets (0 h), the partitioning of αS remained confined predominantly in the periphery of the droplets (Fig. 4b). This is apparent in the enhanced images and their intensity plots where αS can be observed to be localized mainly on the periphery either completely engulfing the droplets (Fig. 4c) or forming multi-phase droplet clusters on the surface as Pickering agents (Fig. 4d-e). Based on the intensities, ~50–60% of the 19% partitioning remained on the surface. FRAP analysis on

the droplets showed an interesting behavior; TDP-43PrLD showed rapid recovery suggesting liquid-like diffusion dynamics. However, αS showed variations in their dynamics depending on the location within the droplets. αS showed liquid-like characteristics within the central core of the droplets based on FRAP recovery albeit mitigated as compared to TDP-43PrLD (Fig. 4f). Interestingly, FRAP from the periphery of the droplets where αS was observed to be preferentially localized, recovery was significantly attenuated (only 20% recovery) indicating a more hardened, gel or solid-like feature (Fig. 4f). To see whether this asymmetric partitioning of αS is time-sensitive, the samples were analyzed after 6 h and 24 h of incubation. After 6 h, the samples looked somewhat identical to the 0 h observation and without any coalescence of droplets which was observed for TDP-43PrLD—RNA droplets in the absence of αS (Fig. S2). After 24 h, droplets in the presence of αS showed diminished coalescence with many distinct droplets maintaining their shapes and sizes (Fig. 4g). This was also apparent in the enhanced images and their intensity plots (Fig. 4h–j). Approximately less than 10% of the droplets showed coalescence, and the initial Pickering effect of αS on the surface slowly changed into engulfing of the droplets by uniformly partitioning on the surface. The engulfment was especially pronounced after 24 h while αS failed to diffuse within the droplet to an appreciable extent. FRAP analysis on the droplets after 24 h significant decrease in the recovery for both TDP-43PrLD and αS (in the center and periphery) suggesting gelation or hardening. To see if there is a conformational bias among αS species for partitioning into the droplets, preformed αS fibrils (αSf) were suspended onto TDP-43PrLD—RNA droplets. Similar to αS monomer incubations, αSf showed Pickering and coarsening effects by remaining on the surface in an aggregated state than TDP-43PrLD for up to 24 h (Fig. S3). However, the engulfment was markedly decreased with αSf as compared to the monomers possibly suggesting a limitation by the conformation of the aggregated state of the protein. A similar experiment with a non-specific protein lysozyme in place of αS as a negative control partitioned completely and colocalized with TDP-43PrLD droplets indicating the selective and unique pattern of αS partitioning (Fig. S4). Together, the data suggest that αS seems to act as a Pickering agent stabilizing and preventing TDP-43PrLD—RNA droplets from coalescing, thereby gelating them.

**Pickering of TDP-43PrLD – RNA liquid droplets by αS is largely driven by electrostatic interactions.** To determine what sequence properties in αS are responsible for the observed Pickering and coarsening effects of TDP-43PrLD – RNA droplets, the αS deletion constructs (αS ΔNTD and αS ΔNAC) were incubated with pre-formed TDP-43PrLD – RNA droplets and were analyzed using confocal microscopy. Samples with αS ΔNTD immediately after incubation showed no signs of Pickering or engulfing as observed with wild-type αS but completely partitioned into the droplets (Fig. 5a) suggesting electrostatics could play a role in asymmetric partitioning, although few αS-studded

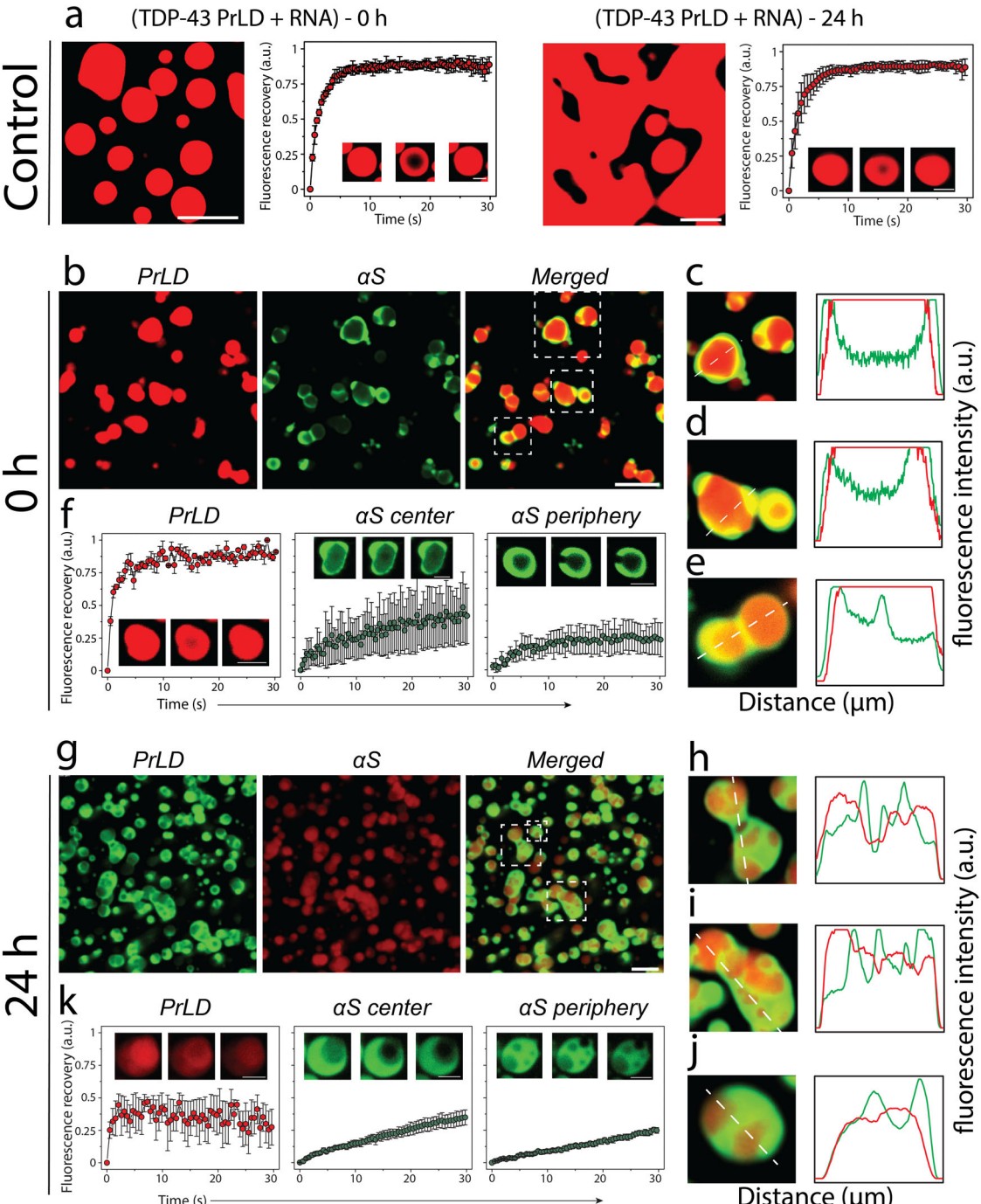

**Fig. 4 Effect of αS on TDP-43PrLD – poly-A RNA condensates. a** Confocal microscopy image showing the phase separation of fluorescently labeled 15 μM TDP-43PrLD in the presence of poly-A RNA at 0 h and 24 h along with their respective fluorescence recovery after photobleaching (FRAP) kinetics. Insets from left to right show pre-bleach, during bleach, and post-bleach of the liquid droplets, respectively. **b** Images were taken right after incubation (0 h) showing the partitioning of αS in the pre-formed TDP-43PrLD—RNA liquid droplets with TDP-43PrLD (red), αS (green), and overlay (yellow). **c** FRAP data on TDP-43PrLD and αS from the samples in (**b**) (n = 3). Images in the inset indicate pre-bleach, during bleach, and post-bleach states, respectively. **d–f** Partitioning analysis of αS based on fluorescence intensity of the representative droplets from (**b**) (white boxes). The intensity plot as a function of distance corresponds to the white dotted line drawn through the droplets. **g–k** Confocal microscopy images and FRAP data of the reactions in (**b, c**) after 24 h along with their respective partitioning analysis (**i, k**). (Scale bar of images = 5 μm, FRAP insets = 2 μm).

droplets were observed (Fig. 5b). FRAP analysis indicated liquid-like characteristics but showed slower recovery kinetics than the wild-type (Fig. 5c, d). After 24 h of incubation, αS's complete partitioning continued to be visible, however, showed an increased number of αS-rich nodes on the surface (Fig. 5i, j). About 50% reduced recovery was observed in FRAP analysis of

αS but TDP-43PrLD recovery did not show any difference from 0 h (Fig. 5k, l). A similar analysis with αS ΔNAC showed different effects on the droplets. Images obtained right after incubation showed that αS predominantly remained on the surface of the droplets (Fig. 5e, f). FRAP data showed that while TDP-43PrLD recovered fully, αS's recovery was significantly mitigated similar

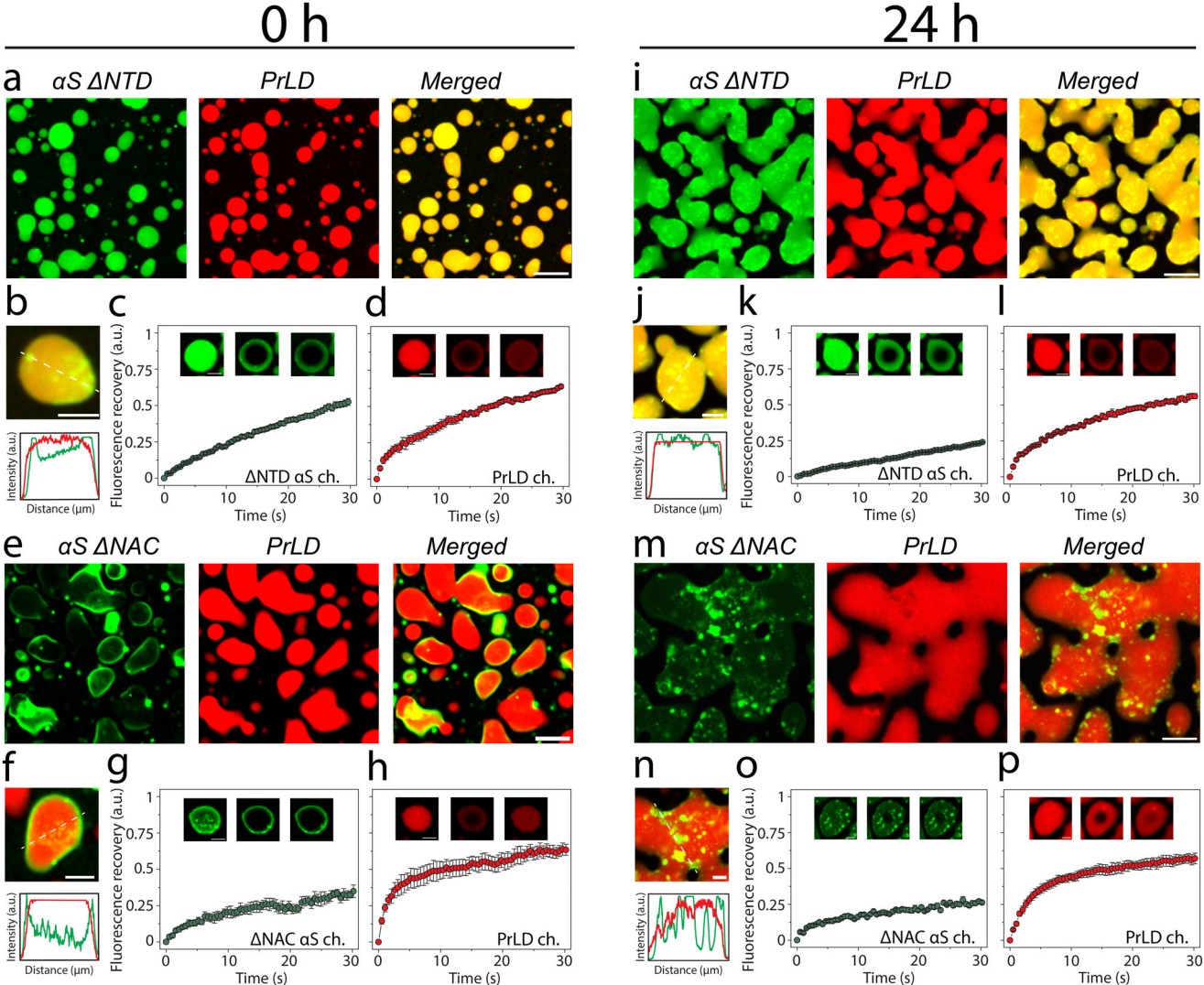

**Fig. 5 Effect of αS domains on the partitioning within TDP-43PrLD – RNA droplets using deletion mutants.** Confocal microscopy images of TDP-43PrLD —RNA preformed droplets immediately upon the addition of αS ΔNTD at 0 hours (**a**), FRAP data showing the recovery of the αS ΔNTD droplets (**b**) and TDP-43PrLD droplets. Insets indicate pre-bleach, bleach, and post-bleach images (**c**). **d** Fluorescence intensity analysis of the merged images of co-incubated TDP-43PrLD—RNA droplets and αS ΔNTD. Intensity corresponds to the white dotted line across the droplets. **e** Partitioning of αS ΔNAC in the preformed droplets of TDP-43PrLD—RNA at 0 h. **f–h** FRAP data showing the recovery of αS ΔNAC (**f**) and TDP-43PrLD (**g**) along with intensity analysis of the merged section of the αS ΔNAC-TDP-43PrLD co-incubated reactions. The same reactions at 0-hour were further monitored after 24 h (**i–p**) to investigate subtle differences in the morphology and internal dynamics of liquid droplets. Scale bar of images = 5 μm, FRAP insets = 2 μm.

to wild-type αS (Fig. 5g, h). After 24 h of incubation (Fig. 5m–p), more widespread coalescence was observed unlike the wild-type suggesting αS ΔNAC could not maintain the emulsifying effect as observed with αS ΔNTD or wild-type. These confocal studies coupled with the quantitative kinetic data suggest that electrostatic interactions could play a role in the Pickering effects and inhibition of coalescence to a greater extent than hydrophobic interactions, which also seem to contribute to some extent.

**αS promotes heterotypic amyloid aggregates of TDP-43PrLD by sequestering RNA to the periphery of TDP-43PrLD – RNA droplets**. To find out if RNA molecules present within the droplets contribute to the asymmetric partitioning of αS, we added the non-specifically fluorescently labeled poly-A RNA and repeated the coincubation of TDP-43PrLD, RNA, and αS. The control TDP-43PrLD—RNA droplets in the absence of αS showed nice spherical droplets in which both TDP-43PrLD and RNA are uniformly dispersed (Fig. 6a). In contrast, TDP-43PrLD—RNA

droplets in the presence of αS showed asymmetric partitioning of both RNA and αS where they were predominantly observed in the surface of the droplets (Fig. 6b). The merged image clearly showed a significant overlap between RNA and αS. These data indicate that RNA is sequestered on the boundary by αS. It is likely that the interaction between αS and RNA is facilitated through αS's charged N-terminal domain. To confirm this, αS ΔNTD was included with TDP-43PrLD-RNA droplets and were observed for RNA sequestration, if any. But as expected, RNA failed to partition within the droplets as αS is devoid of its NTD (Fig. S5). To see whether αS promotes amyloid-like aggregates, the droplets were sedimented by centrifugation and incubated in the presence of thioflavin-T (ThT), a dye that binds to amyloid aggregates and fluoresces. Droplets without αS were treated similarly in parallel as a control. The sample containing αS showed an increase in ThT from 20 hours (red; Fig. 6c) as opposed to the sample without αS, which did not show an ThT increase in intensity even after 50 h (black; Fig. 6c). The

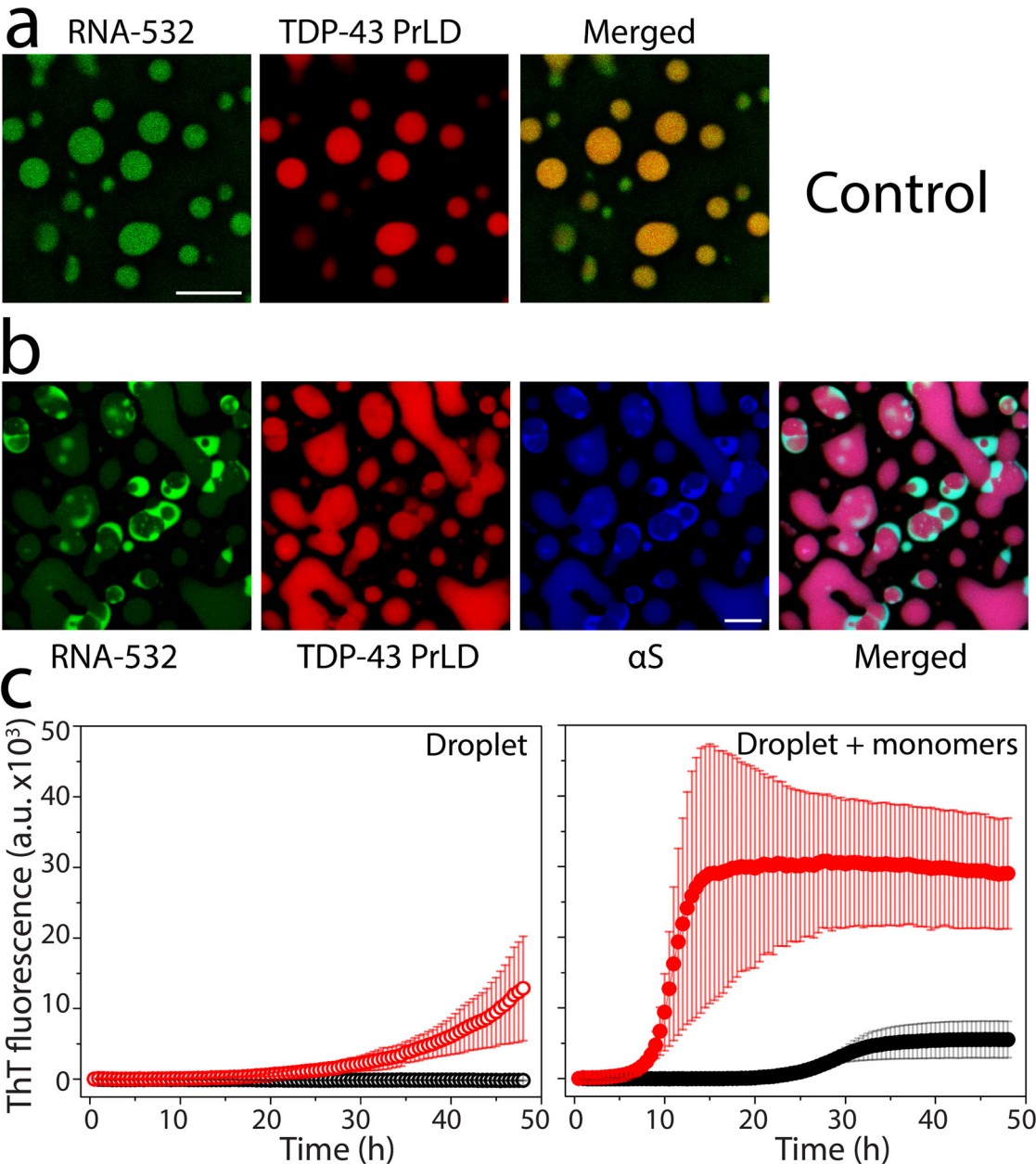

**Fig. 6 RNA sequestration and heterotypic TDP-43PrLD amyloid formation by αS.** TDP-43PrLD and RNA-532 LLPS with and without αS along with the ThT fluorescence of droplets-seeded TDP-43PrLD aggregation. **a**, **b** Confocal microscopic images of TDP-43PrLD droplets formed in the presence of fluorescently labeled RNA (RNA-532). The scale bars in (**a**) and (**b**) are 5 μm. **a** And the partitioning of αS in the preformed TDP-43PrLD and RNA droplets (**b**). **c** ThT fluorescence of control TDP-43PrLD—RNA droplets (black) and TDP-43PrLD—RNA droplets with αS (red) isolated by centrifugation at 18,000 x g (left) and their corresponding ThT fluorescence upon seeding of 5 μM TDP-43PrLD monomers (right). All the experiments were carried out in 20 mM MES buffer pH 6.0 (n = 3).

observed lag time could be due to the combined effects of αS-seeded TDP-43PrLD aggregation as well as the time taken to ThT to permeate into the droplets. Incubation of the sedimented droplets in the presence of 5 μM TDP-43PrLD monomers further exacerbated the lag time difference between the droplets with and without αS; the samples with αS showed a lag time of 8–9 h (red; Fig. 6d) while the ones in its absence showed a lag time of ~25 h (black; Fig. 6d). To further investigate the contributions of RNA on the nucleation and aggregation, a simpler and model in vitro experiment was conducted by incubating RNA with TDP-43PrLD and αS and monitoring the aggregation by ThT fluorescence. Such an experiment showed that only TDP-43PrLD and αS in the presence of RNA showed the shortest lag time vis-à-vis

rapid aggregation as compared to the other controls (Fig. S6a). The presence of RNA in the heterotypic fibrils emerging from the droplets was also confirmed by ethidium bromide staining (Fig. S6b). Together, the data show that αS is able to sequester RNA to the boundary of TDP-43PrLD—RNA droplets and is able to promote ThT-positive amyloid fibrils.

**TDP-43PrLD fibrils seeded by αS-modulated TDP-43PrLD – RNA droplets show a distinctive secondary structure distribution.** To understand the structural modulation of TDP-43PrLD induced by αS containing TDP-43PrLD—RNA droplets, we applied atomic force microscopy (AFM) augmented with infrared (IR) spectroscopy (AFM-IR), which enables simultaneous

measurement of the morphology and secondary structure of the protein aggregates[24–26]. AFM-IR measures the photothermal expansion from resonant IR excitation with an AFM tip, thus allowing for resolving spectral signatures down to individual fibrils[24–28]. This is of pivotal significance in the context of assessing structural changes to a heterogenous ensemble of aggregates, as the spectral facets of the morphologies of interest, can be unequivocally resolved. Furthermore, high spatial resolution allows the identification of the distribution of different secondary structures of proteins within a single fibril[27,28]. The AFM topographic images of control αS and TDP-43PrLD fibrils grown in homogenous solutions exhibit the presence of isolated fibrils (Fig. 7a, c). The respective average IR spectra and corresponding spectral standard deviations in the amide I region obtained from spectra recorded at several spatial locations on the fibrillar aggregates are shown in Fig. 7a, c (Fig. 7b, d). Representative individual spectra and their corresponding spatial locations are shown in Fig. S7. We observe a significant difference in the distribution of secondary structure components between pure αS and TDP-43PrLD fibril controls: the former is more disordered, as evidenced in the relatively high random coil component (1650 cm$^{-1}$ and 1666 cm$^{-1}$), and low antiparallel β-sheet bands (1686 cm$^{-1}$) (Fig. 7b). The spectrum of control pure TDP-43PrLD fibrils generated in homogenous buffer shows increase in the higher wavenumber region and a distinct secondary structure distribution (Fig. 7d). These can be better visualized in the second derivative spectra (green and blue; Fig. 7i). As an additional control, we also acquired spectra from TDP-43PrLD fibrils seeded with TDP-43PrLD-RNA droplets in the absence of αS. The corresponding AFM topograph and spectrum (Fig. 7e, f) showed no major differences in either morphology or the structure of the fibrils from pure TDP-43PrLD control spectrum, indicating that seeding with TDP-43PrLD-RNA droplets does not modulate fibril formation in a significant way (Fig. 7e, f, and yellow; i).

The degree of fibrillar network formation was found to vary across the samples. However, we did not observe morphological variations between fibrillar aggregates, such as twisted and flat morphologies, as typically observed in electron microscopy[29]. We have not attempted to discern between the samples based on the extent of fibrillar network formation. It should be noted that in the presence of fibrillar networks, such as those observed here, the AFM-IR spectral measurements are limited by the sample morphology, and are reflective of fibrillar components of the network, and not isolated, individual fibrils. Nonetheless, this is still a significant improvement over conventional spatially averaged techniques like FTIR, since we can unequivocally attribute the spectral features to fibrillar morphologies and not oligomers or other non-fibrillar aggregates. This approach is valid under the assumption that the spectra from different spatial locations of a fibrillar network, and as an extension, of different components constituting that network, are not significantly different. We have verified this by acquiring AFM-IR spectra from multiple locations, as shown in Fig. S7. Recent developments in mitigating the effects of sample mechanical properties on the AFM-IR[30,31] can enable the acquisition of hyperspectral spatial maps of fibrillar aggregates, which can potentially reveal additional insights into structural variations within networks. Combining AFM-IR with additional high-resolution AFM can also identify subtle morphological variations within a network and their correlation, if any, with secondary structure. We hope to pursue such experimental strategies in future work.

However, this is not the case for TDP-43PrLD fibrils seeded with αS modulated TDP-43PrLD—RNA droplets. While significant variations in fibrillar morphology were not observed (Fig. 7g), the fibril spectra exhibited marked differences and were somewhat similar to pure αS (Fig. 7h and red; 7i). However, it is difficult to draw quantitative structural insights from the second derivative spectra. Therefore, to gain further understanding of the secondary structure distributions in these fibrillar aggregates, we performed spectral deconvolution using the MCR-ALS algorithm[32–35]. The MCR-ALS approach is a blind deconvolution approach that factors spectral data as a weighted linear combination of multiple basis spectra. It is essentially equivalent to a global fit of the spectral data but does not require any a priori knowledge of the component spectra. Additional details on the deconvolution approach are provided in the Methods section. The MCR analysis was performed on a total of 40 spectra, 10 from each fibrillar sample. We find that the spectra can be optimally described as a linear combination of four spectral components, centered at 1626 cm$^{-1}$, 1650 cm$^{-1}$, 1666 cm$^{-1}$, and 1686 cm$^{-1}$. The band at 1626 cm$^{-1}$ can be readily attributed to β-sheet structures, while those at 1650 cm$^{-1}$ and 1666 cm$^{-1}$ likely arise from random coils and β-turns, respectively[36–38] (Fig. 7j). The band at 1686 cm$^{-1}$ is typically attributed to antiparallel β-sheets. The presence of antiparallel structures in αS fibrils has been demonstrated using nonlinear 2D IR spectroscopy[39] and also with AFM-IR[40]. However, the presence of such structures in TDP-43PrLD fibrils has never been investigated. Our results suggest that TDP-43PrLD fibrils concurrently contain both parallel and antiparallel characteristics. Interestingly, similar observations have been recently made on brain-derived fibrillar aggregates of amyloid-β[41,42] and tau fibrils[27]. The weights or concentrations of each of these components for each of the fibril subtypes are shown in Fig. 7k. The error bars represent the standard deviations between the spectra of a given set. The αS-less droplet seeded TDP-43PrLD fibrils exhibit a similar structural distribution as the control TDP-43PrLD fibrils and the only major difference between them is the abundance of antiparallel β-sheets (purple; Fig. 7k). This is an intriguing observation, and suggests that droplet seeding may lead to subtle changes in the β-sheet character of TDP-43PrLD fibrils. For the TDP-43PrLD fibrils seeded from αS modulated droplets, however, we observe a significant shift in the structural distribution from both unseeded pure control and TDP-43PrLD droplet seeded TDP-43PrLD fibrils: the β-sheet populations decrease while the random coil and turn contributions increase (Fig. 7k). This indicates an overall increase in unordered structure in the fibrils in other words, modulation by αS leads to somewhat destabilized fibrils. Taken together, the above analysis unequivocally demonstrates that TDP-43PrLD fibrils seeded with αS modulated droplets are structurally different from pure TDP-43PrLD fibrils.

## Discussion

The connection between SGs and toxic amyloid aggregates remains unclear despite accumulating evidence pointing out that prolonged stress could make SGs become crucibles for amyloid formation[14,15,43]. To understand how SGs could form a nidus for amyloid growth, it is imperative to investigate the effects of heterotypic interactions of amyloid proteins with SGs. We have shown that TDP-43PrLD and αS synergistically interact with each other and form heterotypic amyloid aggregates[16]. With sub-stoichiometric fibril seeds or stoichiometric equivalence of monomers, αS distinctively modulates TDP-43PrLD to induce fibril polymorphs that are distinct in both amyloid core structure and cellular responses[16,17]. These earlier works have established a fundamental framework for the cytoplasmic interactions between the two proteins and sought to answer how heterotypic amyloids could potentiate cellular dysfunction and distinctive phenotypes in pathologies such as LATE, FTLD, and MSA in which colocalized inclusions of TDP-43 and αS have been observed[16,17]. However, one of the key questions is whether cytoplasmic αS

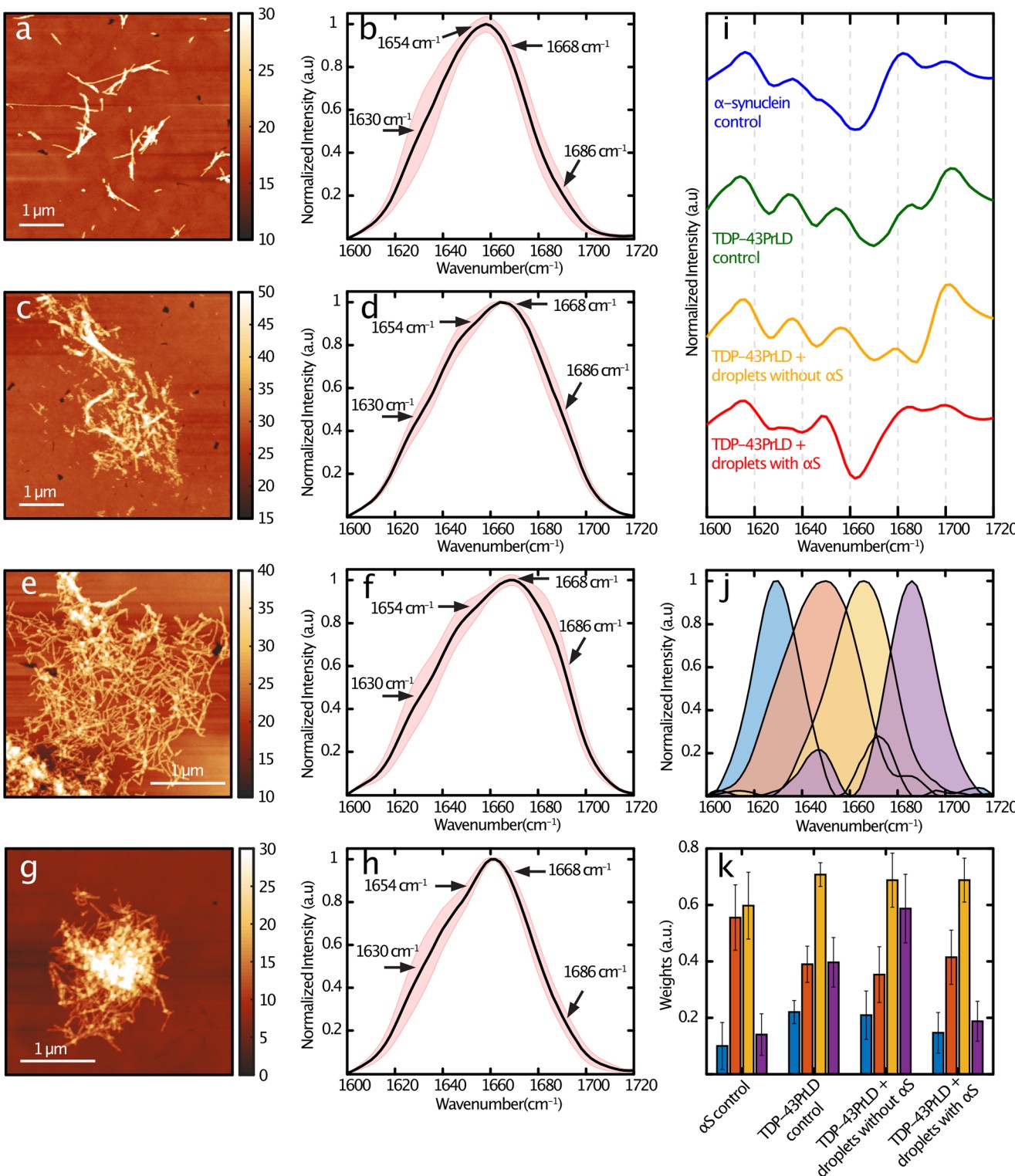

**Fig. 7 AFM-IR spectroscopy of droplets-seeded TDP-43PrLD fibrils.** AFM topographs of fibrillar aggregates of (**a**) αS, (**c**) TDP-43PrLD, (**e**). TDP-43PrLD seeded by TDP-43PrLD-RNA droplets and (**g**). TDP-43PrLD seeded by αS modulated TDP-43PrLD-RNA droplets. **b, d, f, h** Corresponding mean IR spectra. The red-shaded regions represent the spectral standard deviations. **i** Second derivatives of the mean spectra. **j** Spectral components determined from MCR-ALS decomposition. **k** Weights of each of the spectral components for each fibril subtype are color-coded to the corresponding spectral component in (**j**). The error bars represent standard deviations between spectra of a given subtype. The units of the Z-color bar in the AFM images (**a, c, e, g**) is nm.

(monomeric or Lewy body aggregates) can interact with TDP-43 present within the SGs in addition to those present outside these membraneless organelles. Although our preliminary investigations did indicate such a possibility[16], here we have taken a deep

dive into the mechanism of αS interaction with SGs, particularly with TDP-43PrLD – RNA condensates.

Our results demonstrate for the first time that αS interacts with phase-separated TDP-43PrLD—RNA droplets as a Pickering

agent. The phenomenon of Pickering was originally discovered by Ramsden and later by Pickering at the beginning of the 20th century[44,45] and since then Pickering agents have been extensively used to stabilize emulsion preparations in food and pharmaceutical industries[46,47]. 'Pickering emulsions' typically use solid particles as stabilizers of the droplets which partition onto the interface of dense and bulk phases and prevent coalescence. Recently, Folkmann and colleagues showed that the Pickering phenomenon occurs in biomolecular condensates too by demonstrating the clustering of the intrinsically disordered MEG3 protein on the surface of PGL-3 liquid droplets formed in P granules, which leads reduction of surface tension and thereby regulating the exchange between cytoplasm and condensate and maintaining the structural integrity of the droplets[48]. Among several phase-separating amyloid proteins such as tau, TDP-43, and FUS, it has been seen that phase-separated droplets containing these proteins could become foci for amyloid nucleation also[19,49–51]. Recently, the N-terminal domain of prion protein (PrP) was shown to interact with Tau-RNA droplets and promote heterotypic amyloid structures by forming multi-phase clusters somewhat similar to our observations here but did not show Pickering effects[52]. Although some reports have indicated that αS by itself could phase separate into liquid condensates[53], we did not observe the protein phase separate in physiological buffer conditions either by itself or with RNA. In the presence of TDP-43PrLD – RNA droplets, however, αS partitions on to the surface of the droplets as shown in this study.

We observed some interesting aspects in these interactions such as not more than 19% of total αS partitions into TDP-43PrLD – RNA droplets, a majority of which remain on the surface (Fig. 3). Such surprisingly low partitioning of αS in the TDP-43PrLD-RNA droplets suggests that the interactions are driven by specific molecular determinants, interfacial tension and importantly, Pickering effects. The ability of αS to act as a Pickering agent localizing along the droplet's outer surface initially which changes to partial or full engulfment of the droplets, is new and intriguing. A schematic depicting the mechanism of αS based on the results obtained is shown in Fig. 8 which shows the Pickering and engulfment of TDP-43PrLD – RNA droplets. Furthermore, our investigations into the sequence and structural features of αS responsible for the Pickering effects suggest that they are largely driven by electrostatic interactions from the charged residues on the N-terminal domain of αS and to a lesser degree by the central amyloidogenic NAC domain (Fig. 5). Also, the αS monomers seem to be more efficient in Pickering than the fibrils suggesting possibly a conformational selection in αS structure for this phenomenon (Fig. 4 and S3). Similar to the intrinsically disordered MEG-3 in PGL-3 droplets[48], the disordered αS is able to stabilize TDP-43PrLD – RNA droplets by preventing coalescence. With time, Pickering changes into total engulfment of the droplets, which then prevents coalescence. But since αS is aggregation-prone, Pickering and engulfment lead to the hardening of αS coat on TDP-43PrLD – RNA droplets as they age (as observed from negligible FRAP recovery both in vitro and in cells), which likely forms a template for TDP-43PrLD present within the droplets to aggregate as we have previously shown that αS aggregates can efficiently seed TDP-43PrLD monomers[16]. In these interactions, we also sought to understand the role of RNA, which may not be an innocuous spectator. mRNA molecules that assemble on the surfaces of the condensates are also known to stabilize the droplets[54,55]. Along these lines, we also observe that RNA molecules are sequestered to the surfaces of the TDP-43PrLD – RNA droplets upon incubation with αS (Fig. 6). Importantly, RNA seems to contribute to the nucleation of heterotypic aggregates and is present within the fibrils that emerge from it (Fig. 6 and S6). Although we could not quantify the

amount of RNA present in the fibrils, we speculate that RNA engages in non-specific interactions with αS and augments aggregation and therfore, possibly binds to the emerging fibrils in non-stoichiometric proportions. This, in part, could contribute to the structural distinctiveness of the droplet-derived vs. solution-derived heterotypic TDP-43PrLD- αS aggregates. The AFM-IR spectra of isolated fibrillar aggregates revealed that although structural heterogeneity is present in all samples to some degree along a given fibril strand, droplet-derived heterotypic fibrils show distinctly different secondary structure distributions as opposed to the control samples from homogenous solutions (Fig. 7). Although αS mediated TDP-43PrLD fibrils seem to be destabilized, as with our prior observation on heterotypic αS-TDP-43PrLD fibrils generated in homogenous buffers[17], these fibrils can form distinctive polymorphs.

In HeLa cells co-transfected with TDP-43PrLD and αS, cytoplasmic colocalization of the two proteins was observed in non-stressed conditions as seen previously in neuroblastoma cells[17]. Under stress conditions too, αS seems to colocalize with the TDP-43PrLD but based on immunofluorescence with stress markers, this is only a small fraction of cytoplasmic αS colocalized with SGs which could be due to hitchhiking of αS on TDP-43 driven by direct interactions. Interestingly, FRAP analysis indicated almost a full recovery of TDP-43PrLD suggesting a liquid-like state of SGs but the colocalized αS failed to show recovery indicating a gelated/aggregated state. Although we were unable to obtain images at a higher resolution in the cells to see whether αS is localized in the periphery of the SGs (efforts are ongoing), this data parallels the in vitro observation that αS remains in the periphery emulsifying the droplets. Together, the results presented here showcase an intriguing property of αS to interact with TDP-43PrLD—RNA droplets and stress granules as a Pickering agent to emulsify the droplets and nucleate heterotypic amyloid formation. Recently a surprising discovery that αS is able to regulate the homeostasis of RNA processing bodies (P bodies) by binding via the electrostatic N-terminal domain[56] aligns with our discovery that αS could interact and modulate droplets by its N-terminal domain to promote Pickering interactions. Despite known for being associated with numerous functions over three decades, αS continues to surprise everyone. The results presented in this work suggests yet another key modulatory role of αS in biomolecular condensates specifically SG dynamics and TDP-43 proteinopathies adds to the increasing number of roles played by this protein.

## Methods

**Cell culture.** Wild-type HeLa cell lines were a kind gift from Dr. Hao Xu from the University of Southern Mississippi. Cells were cultured in Dulbecco's modified Eagle's medium DMEM (Gibco) supplemented with 10% fetal bovine serum and maintained at 37°C in a humidified incubator with 5.5% $CO_2$. SE Cell Line 4D-Nucleofector™ X Kit (V4XC-1024, Lonza, Germany) was used for transient transfections of plasmids encoding tdTomatoPrLD and Tagged-GFPαS according to the manufacturer's instructions. First, $3 \times 10^5$ cells together with 2 µg plasmid encoding Tagged-GFPαS and 2 µg plasmid encoding tdTomato TDP-43PrLD were resuspended in 80 µL SE cell line solution and transferred to a Nucleocuvette™. The cells were then transfected using the pulse program CN-114 and immediately after supplemented with 500 µL preheated culture medium. Finally, 600 µL cell suspension was transferred to 12-well plate already containing 1 ml preheated culture medium and incubated at 37 °C, 5% $CO_2$ for 35 h. Transfectants were monitored for transient protein expression between 24-35 hours post-transfection. Arsenite treatment was performed after 35 h at a final concentration of 0.5 mM for 20 minutes prior to analysis by confocal microscopy and live cell imaging.

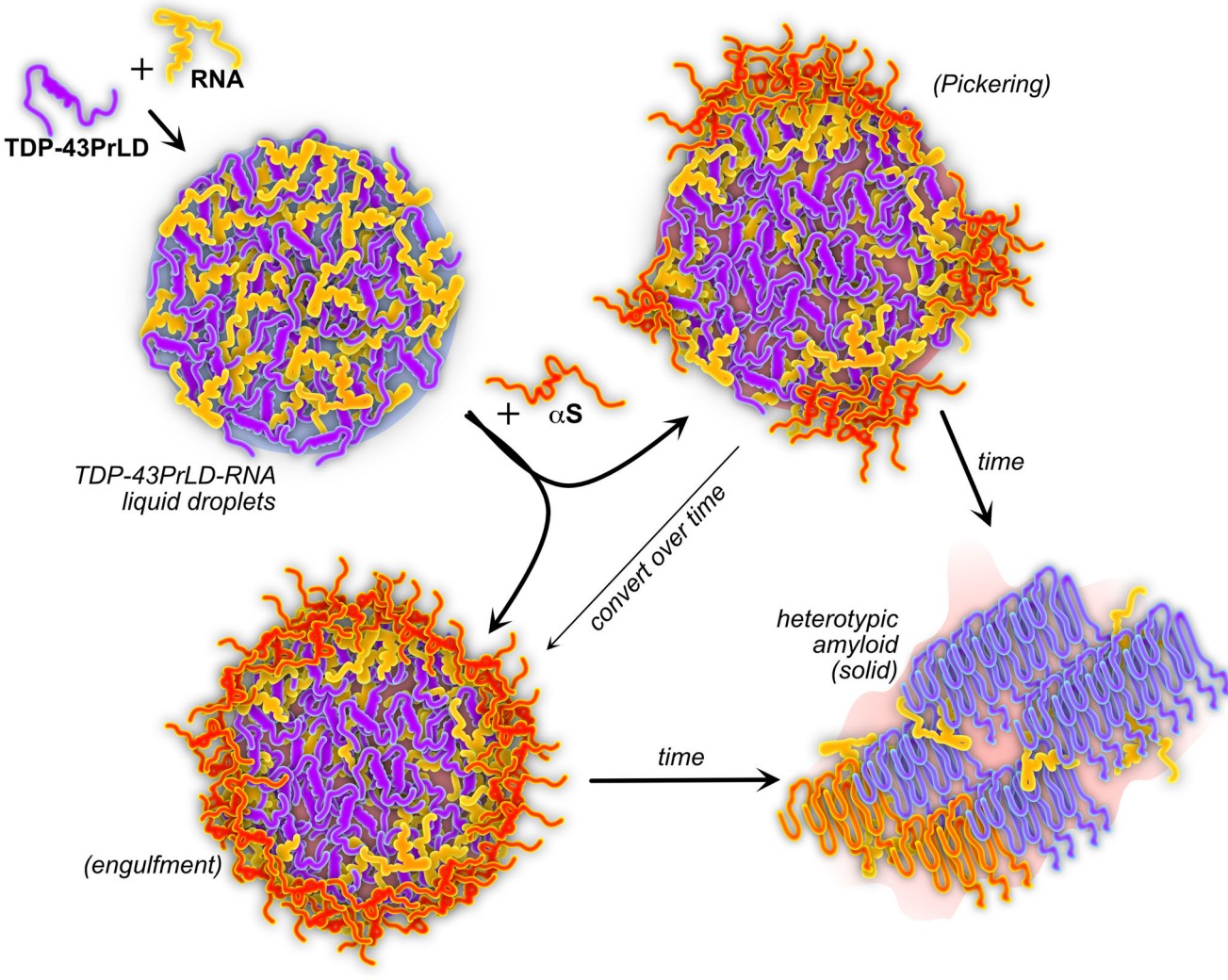

**Fig. 8** Schematic of the mechanism by which αS modulates TDP-43PrLD—RNA droplets.

*i). Immunofluorescence.* Transfectants were seeded on an 18-mm glass coverslip in 12-well black plates (P12-1.5H-N, Cellvis Inc) to reach 60-80% confluency for the next 24 hours. Then, they were fixed with 4% paraformaldehyde in PBS (pH 7.4) after arsenate treatment and permeabilized with 0.2% Triton-X at RT for 10 min. Followed by blocking for 30 min at room temperature with 5% FBS in PBST. The coverslips were then incubated with the TIA-1 antibody (Cell Signaling Technology, TIAR XP rabbit monoclonal antibody #8509); [1:50] in Tris-buffered saline and 0.1% Triton X-100 (TBS-T) containing 5% FBS for overnight at 4 °C, washed with TBS-T three times, and incubated with anti-rabbit secondary antibodies (Cell Signaling Technology, Alexa Flour 488 conjugate #4412) for 1 h at room temperature. After three washes with PBS, the coverslips were mounted on (VEC-TASHIELD® H1900) anti-fade mounting medium (Vektashield) with DAPI and slides visualized on Leica SP8 confocal microscope at 63x magnification. For quantification, four areas of each coverslip were selected, the number of transfected cells and the number of transfected cells with cytoplasmic aggregates were counted, and the ratio was determined per sample.

*ii) In-vivo Confocal microscopy and FRAP.* Live cell FRAP was performed using a 63X objective on cells maintained at 37 °C and supplied with 5% $CO_2$ using an incubator and an objective heater. The cells were incubated with Hoechst33342 for 10 min at 37 °C

for nucleus staining prior to imaging. To induce stress, cells were incubated with 0.5 mM sodium arsenite (Sigma) for 20 minutes after which the cells were imaged. Time lapses were acquired as rapidly as possible over the course of 34 seconds with photobleaching at 554 nm as well as 488 nm FRAP laser for 11 seconds into capture for Tagged-GFPαS and tdTomato TDP-43PrLD respectively. Data were taken from at least different cells ($n = 10$). Regions of interest (ROI) were generated in the photobleached region, a non-photobleached cell, and the background for each time-lapse, and the mean intensity of each were extracted. These values were exported into Origin Pro 8.5 where photobleached and background correction were performed.

*iii) Image processing.* Selected confocal images corresponding to each channel for TDP-43PrLD (red) and αS (green) were transformed into the 16-bit grayscale format and puncta counted using ComDet plugin, and Gaussian blur filter was utilized to enhance puncta visibility. JACoP plugin used for Manders' tM1 and tM2 coefficient calculation per channel. All data was exported and analyzed using OriginPro 8.5 software. All compiled confocal images were processed in Adobe Illustrator and Affinity Designer.

**Recombinant protein expression and purification**
*i). TDP-43PrLD.* TDP-43PrLD plasmid was a kind gift from Dr. Nicolas Fawzi at Brown University (Addgene plasmid

#98669) which was expressed and purified as previously[16,57]. Briefly, the plasmid containing TDP-43PrLD as a fusion construct with an N-terminal hexahistidine tag and with a tobacco etch virus (TEV) protease cleavage site was transformed in *E. coli* BL21 (DE3) star cells (Invitrogen) cells. The cells were cultured overnight in LB medium containing 100 µg/mL kanamycin at 37 °C. The following morning, the culture was transferred to a fresh media and grown until the $OD_{600}$ reaches 0.6–0.8, followed by inducing the expression using 1 mM isopropyl 1-thio-β-D galactopyranoside (IPTG). The cells were transferred to an incubator shaker at room temperature (25 °C, 220 rpm) and grown overnight. Cells were then harvested by centrifugation (12000 rpm for 15 minutes at 4 °C) and the pellet was stored at −80 °C until further use. For protein purification, the cells were resuspended in lysis buffer [20 mM Tris (pH:8.0), 500 mM NaCl, 5 mM imidazole and 6 M Urea) followed by the addition of 0.5 mM PMSF and were lysed with Misonix XL-2000 sonicator. The lysate was centrifuged at 20,000 rpm at 4 °C for 45 minutes and the supernatant was resuspended in Kimble® KONTES® Flex column containing Ni(II) NTA resin and incubated for 1 h. It was followed by two washing steps with the wash buffers, W1 and W2 containing 20 mM Tris, 500 mM NaCl, and 6 M urea at pH 8.0 containing 15 and 30 mM imidazole, respectively, and the protein was eluted in elution buffer (20 mM Tris, 500 mM NaCl, 6 M urea at pH 8.0 containing 150 mM imidazole). The protein was then dialyzed against 20 mM Tris, 500 mM NaCl, 2 M urea, at pH 8.0, concentrated, and stored at −80 °C until further use. Before use, the protein was desalted using Sephadex G-25 HiTrap desalting column (Cytiva) in 20 mM MES buffer pH 6.0 and its purity was confirmed using MALDI-ToF as previously[58].

*ii). Wild-type αS, αS ΔNTD, and αS ΔNAC.* Wild-type plasmids were synthesized from Genscript ® and the rest of the constructs were cloned at the molecular cloning facility at Florida State University. All proteins were expressed in *E. coli* BL21 (DE3) star cells (Invitrogen) as a fusion construct with an N-terminal His-tag (His6) followed by thrombin cleavage site and purified as previously[17,59]. Briefly, the cells were resuspended in the lysis buffer (20 mM Tris, 100 mM NaCl, pH 8.0) and lysed by soni-cation. The lysate was centrifuged at 20,000 rpm for 45 minutes at 4 °C. The supernatant was incubated with the HisPur™ Ni(II) NTA beads, incubated and washed twice with wash buffer, the first wash containing 20 mM and the second wash 40 mM imi-dazole, and the protein was eluted in elution buffer (20 mM Tris 100 mM NaCl, and 300 mM imidazole at pH 8.0). The protein was concentrated using Amicon filters and subjected to size exclusion chromatography to remove any αS preformed aggre-gates. These samples were immediately used after buffer exchange in 20 mM MES pH 6.0.

**Preparation of RNA and labeling**. Lyophilized powdered poly-A RNA was acquired from Sigma and dissolved in RNase-free water. The concentration was determined using the conversion factor of 1 absorbance equivalent to 40 µg/mL. The prepared stock was frozen and stored at −80 °C. Prepared stock aliquots were thawed and used for the experiments immediately. For labeling, Bactoview™ Live Green (500x, Biotium) was directly added to the RNA stock prior to incubating with the reaction at 1X final concentration.

**Labeling of monomeric αS and TDP-43PrLD**. αS and TDP-43PrLD were fluorescently labeled using Hilyte Flour 532/647 succinimidyl ester (Anaspec) and Hilyte Flour 647, respec-tively. Proteins were incubated for 16 hours at 4 °C with a 3 M excess of dye in 20 mM MES solution pH 6.0. TDP-43PrLD and

αS were eluted in sterile 20 mM MES buffer pH 6.0 using two cycles of the Sephadex G-25 Spin Trap desalting column (Cytiva). For confocal imaging, TDP-43PrLd fluorescently labeled using Hilyte Flour 647 succinimidyl ester (Anaspec) and αS labeled with Hilyte Flour 532 succinimidyl ester (Anaspec) were used at 1–2% of final protein concentration.

**Quantitative kinetics assay**. For quantitative kinetics analysis, droplets of TDP-43PrLD – RNA were formed by incubating 10 µM of TDP-43PrLD in 20 mM MES buffer at pH 6.0 with 50 µg/mL RNA for 10 minutes at room temperature. To this, αS monomers, 10% of which were labeled with HiLyte 647 dye, were added. At specific time points, the samples were sedimented by centrifugation for 20 minutes at 19,000xg, and the supernatant was measured for the concentration of αS based on absorbance at 650 nm using Cary 300 Varian UV spectrophotometer. Data was processed using Origin 8.5 for at least three independent measurements.

**Turbidity assay**. Turbidity was measured at 600 nm using a BioTek Synergy H1 microplate reader. Prior to each measure-ment, reactions were equilibrated at room temperature for 10 minutes. A boundary value of 1.40 optical density 600 ($OD_{600}$) was selected for the purpose of establishing phase diagrams. Temperature scans on phase diagrams were made by mixing the samples and equilibrating them for 15 minutes at each tempera-ture. The data were processed using Origin 8.5, by averaging at least three different datasets.

**Thioflavin-T fluorescence**. Thioflavin-T (ThT) fluorescence assays were measured on a BioteK Synergy H1 microplate reader by adding 10 µM ThT to the samples probed. The samples were excited at 452 nm and the emission at 485 nm was monitored at 37 °C.

**Confocal microscopy, FRAP, and intensity analysis**. Confocal microscopic images of the TDP-43PrLD droplets with and without αS were acquired using Leica STELLARIS-DMI8 microscope at 100X magnification in a 96-well clear bottom, black plates (P96-1.5H-N, Cellvis Inc.). All the proteins used in the confocal experiments were resuspended in 20 mM MES buffer pH 6.0 and contained up to 1-2% of the fluorescently labeled protein. In all reactions, TDP-43PrLD and RNA droplets were allowed to settle for a few minutes and αS was gently added from the top without disturbing the settled condensates at the bottom of the plate, immediately followed by imaging. Fluorescence recovery after photobleaching (FRAP) was employed to study the internal dynamics of the TDP-43PrLD and RNA condensates with or without αS. The liquid droplets were photobleached with a laser intensity of ~90% for 10 s and the fluorescence recovery was monitored for 30 s. The kinetics of fluorescence recovery were normalized and plotted against time using Origin 8.5. Finally, the fluorescence intensity profile across the selected region of the droplet was plotted using the ImageJ software.

**AFM-IR**. AFM-IR experiments were carried out on a Bruker NanoIR3 (Bruker corporations) instrument equipped with a mid-IR quantum cascade laser (MIRcat, Daylight solutions). All experiments were performed at room temperature and low rela-tive humidity (<5%). AFM images and IR spectra were recorded in tapping mode with cantilevers having a resonance frequency of 75 ± 15 kHz and spring constant of 1-7 N/m. IR spectral resolu-tion was kept at 2 cm$^{-1}$ and a total of 128 co-additions at every point with 64 co-averages were applied for each spectrum. AFM images were processed with Gwyddion software and IR spectra

were processed with MATLAB software. A (3,7) Savitzky-Golay filter and a 3-point moving average filter were used to denoise every spectrum.

**Spectral deconvolution using MCR-ALS**. The MCR-ALS algorithm, implemented in MATLAB by Jaumot et al.[32,33], was used for spectral deconvolution. A total of 40 spectra (10 from each fibril type) was used for the deconvolution. MCR-ALS essentially is a matrix factorization approach that determines the pure spectral responses (S) and their corresponding weights/concentrations (C) from a spectral dataset D as: $D = C*T$[32-35]. The number of spectral components was chosen to be 4, which is consistent with the number of peaks observed in the second derivative spectra. Four Gaussian bands, centered at 1626 cm$^{-1}$, 1646 cm$^{-1}$, 1666 cm$^{-1}$, and 1690 cm$^{-1}$ (obtained from the second derivative of the mean spectra) were used as initial spectral estimates for the MCR-ALS algorithm.

**Reporting summary**. Further information on research design is available in the Nature Portfolio Reporting Summary linked to this article.

## Data availability

All data supporting the findings of this study are available within the paper and its Supplementary Information. The source files for all the graphs presented in the paper are also available in FigShare[60].

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

## Acknowledgements
We would like to thank the National Science Foundation (BMAT 2208349 to VR) and the National Institute of General Medical Sciences (R35 GM138162 to AG) for their financial support. Also, the authors thank the National Center for Research Resources (5P20RR01647-11) and the National Institute of General Medical Sciences (8 P20 GM103476-11) from the National Institutes of Health for funding through INBRE for the use of their core facilities. The authors would also like to thank the INBRE facility manager, Dr. Jonathan Lindner for his gracious help in using the confocal microscope. The authors also acknowledge Dr. Nicholas Fawzi for donating plasmids.

## Author contributions
V.R. conceptualized the project; S.D. and M.M. performed protein expression and biophysical experiments, A.M. performed the cell culture experiments; S.B. and A.G. collected and analyzed AFM-IR data; V.R., S.D., S.B., and A.G. participated in intellectual discussions and manuscript writing and editing.

## Competing interests
The authors declare no competing interests.
