## [Peer Review File · Communications Biology]

Reviewers' comments:

Reviewer #1 (Remarks to the Author):

In this manuscript by Dhakal et al. titled " α -Synuclein emulsifies TDP-43 prion-like domain-RNA liquid droplets to promote heterotypic amyloid fibrils", the authors beautifully demonstrate the Pickering effect of α -synuclein on the heterotypic TDP-43 PrLD-RNA droplets and further liquid-to-solid maturation into amyloid aggregates. The authors show cellular co-localization of TDP-43 PrLD and α -synuclein under normal and stressed conditions within cytoplasmic puncta and stress granules. The authors further monitor the partitioning of α -synuclein with the complex coacervates formed by TDP-43 PrLD and RNA by varying the protein-RNA stoichiometry and temperature, with different α -synuclein truncation mutants. By confocal imaging, the authors show the preferential recruitment of α -synuclein monomers and aggregates along with RNA on the droplet periphery leading to the emulsifying effect on the heterotypic TDP-43 PrLD-RNA droplets. This beautiful work illustrates the role of electrostatic interactions in driving the sequence-determined distinctive partitioning behavior of wild-type and variant α -synuclein monomers within the complex TDP-43 PrLD-RNA droplets. Using AFM-IR, the authors also characterize the structure of the amyloid aggregates formed in the absence and presence of droplet seeds of TDP-43 PrLD-RNA droplets along with α -synuclein monomers. This study would be of great interest to the readership of this journal. I have some suggestions that the authors can consider incorporating in the revised manuscript.

1. In Figures 1a and 1c, labeling in the figure (stress and non-stress conditions) and legends do not match.

2. In Figure 1c, the cell nucleus is not visible within the cell. Have the authors not included the DAPI channel in the merged image as opposed to the image in 1a? The authors can consider replacing the current representative images with images that better visualize the cytoplasmic puncta and co-localization of the proteins under study.

3. Can the authors discuss the possible reasons behind the reduced FRAP recovery within puncta formed in non-stress conditions as compared to the SGs? Can the authors quantify the differential recruitment of TDP-43-PrLD and α -synuclein in normal (non-stress) and stressed conditions?

4. On Page 8, Fig. 3g should be Fig. 3c.

5. On Page 8, it should be 25 and 37°C as displayed in Fig. 3e.

6. In Figure 4g, the images of the green (α -synuclein) and red (TDP-43 PrLD) channels seem to be reversed.

7. The authors suggest that electrostatic interactions play an important role in exhibiting the Pickering effect of α -synuclein on the TDP-43 PrLD-RNA droplets, which is well supported by the experiments performed with the deletion mutants of α -synuclein. However, as RNA is also known to interact electrostatically with proteins and exhibits preferential surface localization in the presence of α -synuclein, it'd be interesting to observe the distribution of RNA within TDP-43 PrLD-RNA condensates in the presence of deletion mutants of α -synuclein.

8. A thorough proofreading to fix several typos/grammatical errors in some sentences is required.

Reviewer #2 (Remarks to the Author):

The authors elucidated the physical properties of the localization and fluidity within stress granule

droplets formed by TDP-43 and α -synuclein, as well as how the secondary structure of fibrous aggregates is regulated by α -synuclein, using various biochemical and biophysical methods. In particular, it was found that α S localizes on the surface of SGs and prevents the coalescence of droplets, which is a novel aspect of the study from the perspective of Pickering emulsification. This research offers a new insight into how the physical properties of droplets formed inside cells are regulated by the localization of proteins, making it intriguing. On the other hand, there are some significant flaws observed. Firstly, there is insufficient research and consideration regarding the physical mechanism of how the Pickering effect of α S affects the formation of heterotypic fibril aggregates. It is unclear why the localization of α S on the droplet surface is necessary for the formation of heterotypic fibers. Furthermore, it's unknown whether the formation of heterotypic fibers is crucial for the pathogenesis of neurodegenerative diseases. The second concern is that the validation of whether heterotypic fibrils are genuinely formed is inadequate. The problem lies in the verification method with AFM-IR. The fibers of droplets containing both TDP-43rLD and α S are quite entangled, raising doubts about whether the results truly reflect the secondary structure of a single fiber. Below are specific concerns.

Major points

1. Page 9, "Equimolar incubations with 10 μ M α S and TDP-43PrLD showed only 19% of total α S partitioning in the droplets (●; Fig. 3g), while sub-stoichiometric (5 μ M) and excess (20 μ M) α S showed 17% (● and ●; Fig. 3g) and were statistically insignificant from one another. The author should mention the physical factors why α S is not incorporated into the droplet even though there is an excess amount of α S.
2. Page 9, "Based on these results that show the N terminal domain affecting the Pickering effect more than the central hydrophobic domain suggests it could be largely an electrostatically-driven process." It is strange that the topic of Pickering suddenly appears when it hasn't been mentioned up to this point. At this stage, no data has been presented about the localization of α S in droplets, so it's odd to mention its impact on Pickering. Also, in Fig. 3i, Δ NTD seems to incorporate α S into the droplets faster and in larger quantities than Δ NAC. That is, the effect of Δ NAC (hydrophobic effect) seems to be greater. Isn't this the opposite of what is being mentioned here?
3. Page 11, "The surprising observation that only 19% of the total α S partitions into TDP-43PrLD – RNA droplets prompted us to investigate the partitioning behavior by confocal microscopy." The reviewer doesn't quite grasp why it's surprising that only 19% of α S is incorporated into the droplets. Could the authors explain in more detail?
4. Page 11, "This was also apparent in the enhanced images and their intensity plots that continued to show Pickering effects (Fig. 4h-j). " From as far as I see Fig. 4h-j, it appears that α S has penetrated considerably into the interior of the droplets. I'm doubtful if this can still be said to be a result of the Pickering.
5. Page 11, "However, the 'engulfment' of the droplets was markedly decreased with α Sf as compared to the monomers possibly suggesting a limitation by the aggregated state of the protein for partitioning" I'm not sure what the term 'engulfment' refers to. It should be explained in the confocal microscopy figure, perhaps with arrows indicating what is being referred to.
6. Page 13, "These confocal studies coupled with the quantitative kinetic data suggest that electrostatic interactions could play a role in Pickering and inhibition of coalescence to a greater extent than hydrophobic interactions." As the authors have mentioned, the droplets coalesced 24 hours later. Therefore, the conclusion that electrostatic interactions contribute to the 'inhibition of coalescence' sounds questionable.
7. Page 15, "The sample containing α S showed an increase in ThT from 20 hours (i; Fig. 6c) as

opposed to the sample without α S which did not show an increase in ThT intensity even after 50 hours (i; Fig. 6c). Incubation of the sedimented droplets in the presence of 5 μ M TDP-43PrLD monomers further exacerbated the lag time difference between the droplets with and without α S. The samples with α S showed a lag time of 8-9 hours (l; Fig. 6d) while the ones in its absence showed a lag time of \sim 25 hours (l; Fig. 6d)."

Why is ThT added to the sedimented droplets? Wouldn't it be better to observe ThT at the same time as droplet formation? I think fibrils may already be forming inside the droplets, so why can't ThT fluorescence be observed until 20 hours have passed? Could it be the time it takes for ThT to permeate the droplets? Also, how does the increase in the ThT S-curve with added monomer relate to the droplets? Are fibrils forming from monomers on the droplet surface? If α S is added to the solution without droplets, wouldn't such an increase in ThT fluorescence be observed?

8. Page 15, "The presence of RNA in the heterotypic fibrils emerging from the droplets was also confirmed by ethidium bromide staining (Fig S5b)."

How can the fibrils be determined to be heterotypic? Evidence should be provided.

9. Page 17, "...as evidenced in the relatively high random coil component (1650 cm^{-1} and 1666 cm^{-1}), and low antiparallel β -sheet bands (1686 cm^{-1}) (Fig. 7b)."

The reviewer cannot recognize relationship between the wavenumber and IR spectral features claimed by the author on Fig. 7b, and even Fig. 7i. The features should be indicated by arrows or other means in the figure.

10. Page 17, "While significant variations in fibrillar morphology were not observed (Fig. 7g), the fibril spectra exhibited marked differences, and were somewhat similar to pure α S (Fig. 7h and red; 7i)." Figs. 7e and g appear to have very different degrees of agglomeration. This makes it a bit of a stretch to claim that there is no significant difference.

11. Since Figure 7g shows a large aggregation of fibrils, the AFM probe should not detect the vibration from only a single fibril directly under the probe. So, if some fibrils were overlapped, wouldn't the IR spectrum be a mixture of both effects? Thus the reviewer considers what can be said about secondary structure from this experiment is rather uncertain. If the author could map the IR spectrum peaks at specific wavelengths, they would be able to show how much the structure differs from fibril to fibril and inhomogeneity of fibrils. It might also be possible to show differences in secondary structure by showing a slightly more enlarged, higher resolution AFM image to show differences in fibril morphology.

Minor points

1. Page 4, "Our results, obtained from a combination of high-resolution microscopy and "high-resolution microscopy" should be "Confocal optical microscopy". Moreover, for 'nanoscale, spatially-resolved AFM-IR spectroscopy', the term 'spatially-resolved' in particular feels out of place since it's not doing mapping. It should simply be "AFM-IR spectroscopy'."

2. Page 6, "...gelation (■; Fig. 1c). In contrast, α S showed no recovery in these puncta which suggests that α S could be in a more gelated, aggregated state (■; Fig. 1c)." "Fig.1c" should be "Fig. 1b".

3. Page 8, "... μ g/mL in the absence of α S (●; Fig. 3g)."

Color of the solid circles in Fig. 3 is inconsistent between the main text, the figure captions, and the graph.

4. Page 8, "...above and below UCST (4 and 37 °C) showed no changes..)" "4" should be "25".

5. Page 13, "To determine what sequence properties in α S are responsible for the observed Pickering and coarsening effects of TDP-43PrLD – RNA droplets, truncation constructs α S Δ NTD and α S Δ NAC were designed. To broadly identify the effects of two important domains of α S; the N-terminal domain (NTD; 1-60) is highly charged and the central amyloid region (NAC; 61-95) is rich in hydrophobic residues and therefore, deletion of these regions could provide clues for their contributions in Pickering.

The data for the deletion mutant is already presented in Fig. 3i. Therefore, this explanation should be given before referring to Fig 3i.

6. Fig 7 a, d, e, g

The Z color bar should be marked with a unit.

Reviewer #3 (Remarks to the Author):

This is an interesting and important study dedicated to the in-depth analysis of the peculiarities of the interactions between TDP-43PrLD and alpha-synuclein using cell culture models and in vitro TDP-43PrLD-RNA liquid droplets. The authors showed that the alpha-synuclein forms clusters on the surface of TDP-43PrLD – RNA droplets and emulsifies them. They also showed that such alpha-synuclein-driven emulsification nucleates formation of the heterotypic amyloids. Such heterotypic amyloids are characterized by specific structural features, which are different from those of fibrils derived from the homogeneous solutions. The authors hypothesized that in this system, alpha-synuclein serves as a Pickering agent leading to the formation of a Ramsden or Pickering emulsion; i.e., emulsion stabilized by solid particles locating to the droplet surface.

This work adds significantly to the field and will have a noticeable impact. Although the manuscript is generally well-written and concise, it has some linguistic issues and requires careful editing and proofreading.

Reviewer #4 (Remarks to the Author):

In this manuscript by Dhakal et al. titled " α -Synuclein emulsifies TDP-43 prion-like domain–RNA liquid droplets to promote heterotypic amyloid fibrils", the authors beautifully demonstrate the Pickering effect of α -synuclein on the heterotypic TDP-43 PrLD-RNA droplets and further liquid-to-solid maturation into amyloid aggregates. The authors show cellular co-localization of TDP-43 PrLD and α -synuclein under normal and stressed conditions within cytoplasmic puncta and stress granules. The authors further monitor the partitioning of α -synuclein with the complex coacervates formed by TDP-43 PrLD and RNA by varying the protein-RNA stoichiometry and temperature, with different α -synuclein truncation mutants. By confocal imaging, the authors show the preferential recruitment of α -synuclein monomers and aggregates along with RNA on the droplet periphery leading to the emulsifying effect on the heterotypic TDP-43 PrLD-RNA droplets. This beautiful work illustrates the role of electrostatic interactions in driving the sequence-determined distinctive partitioning behavior of wild-type and variant α -synuclein monomers within the complex TDP-43 PrLD-RNA droplets. Using AFM-IR, the authors also characterize the structure of the amyloid aggregates formed in the absence and presence of droplet seeds of TDP-43 PrLD-RNA droplets along with α -synuclein monomers. This study would be of great interest to the readership of this journal. I have some suggestions that the authors can consider incorporating in the revised manuscript.

1. In Figures 1a and 1c, labeling in the figure (stress and non-stress conditions) and legends do not match.

2. In Figure 1c, the cell nucleus is not visible within the cell. Have the authors not included the DAPI channel in the merged image as opposed to the image in 1a? The authors can consider replacing the current representative images with images that better visualize the cytoplasmic puncta and co-localization of the proteins under study.
3. Can the authors discuss the possible reasons behind the reduced FRAP recovery within puncta formed in non-stress conditions as compared to the SGs? Can the authors quantify the differential recruitment of TDP-43-PrLD and α -synuclein in normal (non-stress) and stressed conditions?
4. On Page 8, Fig. 3g should be Fig. 3c.
5. On Page 8, it should be 25 and 37°C as displayed in Fig. 3e.
6. In Figure 4g, the images of the green (α -synuclein) and red (TDP-43 PrLD) channels seem to be reversed.
7. The authors suggest that electrostatic interactions play an important role in exhibiting the Pickering effect of α -synuclein on the TDP-43 PrLD-RNA droplets, which is well supported by the experiments performed with the deletion mutants of α -synuclein. However, as RNA is also known to interact electrostatically with proteins and exhibits preferential surface localization in the presence of α -synuclein, it'd be interesting to observe the distribution of RNA within TDP-43 PrLD-RNA condensates in the presence of deletion mutants of α -synuclein.
8. A thorough proofreading to fix several typos/grammatical errors in some sentences is required.

Reviewer #1

1. In Figures 1a and 1c, labeling in the figure (stress and non-stress conditions) and legends do not match.

Response: We have now corrected and updated the figure legend in the main text.

2. In Figure 1c, the cell nucleus is not visible within the cell. Have the authors not included the DAPI channel in the merged image as opposed to the image in 1a? The authors can consider replacing the current representative images with images that better visualize the cytoplasmic puncta and co-localization of the proteins under study.

Response: Thank you for pointing this. We also realized that DAPI channel was missing in Figure 1c and has now been corrected.

3. Can the authors discuss the possible reasons behind the reduced FRAP recovery within puncta formed in non-stress conditions as compared to the SGs? Can the authors quantify the differential recruitment of TDP-43-PrLD and α -synuclein in normal (non-stress) and stressed conditions?

Response: Thank you for this excellent point. We want to bring to the reviewer's attention that we have previously demonstrated that TDP-43PrLD and α S synergistically interact and form heterotypic fibrils in vitro [1]. This interaction was also captured in SH-SY5Y cells where both proteins formed cytoplasmic puncta under non-stress conditions [2], which supported our in vitro data as being heterotypic aggregates. Based on these prior results and attenuated FRAP recovery observed in both α S and PrLD in non-stress situations, suggests that these are puncta of heterotypic aggregates. It is also important to note that under non-stress conditions, no stress granules are present. We have now modified our statement in the manuscript. "*These results suggest that while under non-stressed conditions when no SGs are present, α S and TDP-43PrLD co-localize as puncta in the cytosol possibly as heterotypic aggregates as demonstrated earlier [2], under stress conditions, α S colocalizes with TDP-43PrLD present in SGs*". With regards to the quantification, unfortunately, we are unable to directly compare the amounts of TDP-43PrLD forming heterotypic aggregates with α S under non-stress conditions and the interaction of the two in stress granules. This is mainly because the two may not be mutually exclusive and competing processes, and the fate of the two proteins may also depend on their presence in other organelles and membranes.

4. On Page 8, Fig. 3g should be Fig. 3c.

Response: Fixed.

5. On Page 8, it should be 25 and 37°C as displayed in Fig. 3e.

Response: Fixed.

6. In Figure 4g, the images of the green (α -synuclein) and red (TDP-43 PrLD) channels seem to be reversed.

Response: Fixed

7. The authors suggest that electrostatic interactions play an important role in exhibiting the Pickering effect of α -synuclein on the TDP-43PrLD-RNA droplets, which is well supported by the experiments performed with the deletion mutants of α -synuclein. However, as RNA is also known to interact electrostatically with proteins and exhibits preferential surface localization in the presence of α -synuclein, it'd be interesting to observe the distribution of RNA within TDP-43 PrLD-RNA condensates in the presence of deletion mutants of α -synuclein.

Response: Thank you for the great suggestion. We anticipated that since α S Δ NTD abrogated Pickering effects and diffused inside TDP-43PrLD droplets, we may not observe RNA sequestration. However, based on this suggestion, we conducted the experiment with the mutant as anticipated, no sequestration was observed further confirming that the sequestration was due to the electrostatic interaction of α S NTD

with RNA. We have now added this figure in supplementary data (see below) and the conclusions have been added to the results section: *“It is likely that the interaction between αS and RNA is facilitated through αS 's charged N-terminal domain. To confirm this, $\alpha S \Delta NTD$ was included with TDP-43PrLD-RNA droplets and were observed for RNA sequestration, if any. As expected, RNA failed to partition within the droplets as αS is devoid of its NTD (Fig S5).”*

8. A thorough proofreading to fix several typos/grammatical errors in some sentences is required.

Response: We have now done proofreading to fix some of these errors.

Reviewer #2

Major points

1. Page 9, *“Equimolar incubations with 10 μM αS and TDP-43PrLD showed only 19% of total αS partitioning in the droplets (●; Fig. 3g), while sub-stoichiometric (5 μM) and excess (20 μM) αS showed 17% (● and ●; Fig. 3g) and were statistically insignificant from one another.”* The author should mention the physical factors why αS is not incorporated into the droplet even though there is an excess amount of αS .

Response: We thank the reviewer for this question. When αS is added to the preformed TDP-43PrLD droplets, the protein partitions in the droplets; in our kinetic assays (Fig 3f-i), we were unable to distinguish between complete partitioning within, as well as interaction on the surface. Therefore, the total 19% measured is the sum of those that are fully partitioned inside and those that remain on the surface. We believe the reason behind the total amount partitioning being constant (17-19%) irrespective of the stoichiometry is due to αS remaining on the surface of the TDP-43PrLD – RNA droplets and aggregating, which prevents further partitioning of αS from the bulk phase, as was confirmed by subsequent experiments. These experiments also confirmed that electrostatic forces are largely responsible for the Pickering effects. To clarify this point further, the following statement has been incorporated in the manuscript; *“The partitioning of αS not showing significant quantitative change despite varying stoichiometry could be attributed to the protein’s preferential localization on the surface of preformed TDP-43PrLD droplets preventing further recruitment from the bulk/dilute phase.”*

2. Page 9, *“Based on these results that show the N terminal domain affecting the Pickering effect more than the central hydrophobic domain suggests it could be largely an electrostatically-driven process.”* It is strange that the topic of Pickering to suddenly appear when it hasn't been mentioned up to this point. At this stage, no data has been presented about the localization of αS in droplets, so it's odd to mention its impact on Pickering. Also, in Fig.3i, ΔNTD seems to incorporate αS into the droplets faster and in larger quantities than ΔNAC . That is, the effect of ΔNAC (hydrophobic effect) seems to be greater. Isn't this the opposite of what is being mentioned here?

Response: This is an excellent point and thank the reviewer for bringing this up. We have now removed the statement on the Pickering effect that was prematurely presented. Instead, we conclude the description of Figure 3 with this statement: *“These results suggest that both hydrophobic and electrostatic interactions differentially contribute to the partitioning of αS to the TDP-43PrLD – RNA droplets.”* We also understand the reviewer’s concern regarding Figure 3i. We would like to clarify that although there is a kinetic rate differences between $\alpha S \Delta NTD$ and $\alpha S \Delta NAC$ as we describe in the manuscript, the amount

incorporated largely remains the same (please see the 20h data points in Fig 3i). Only the rate of partitioning is low. Furthermore, we want to bring to the reviewer's attention that Pickering and partitioning are distinct and different processes. While $\alpha\Delta\text{NTD}$ significantly attenuates the Pickering effect (Fig 5a, i), it is observed with $\alpha\Delta\text{NAC}$ (Fig 5e,m), although it does not prevent coalescence suggesting there could be some hydrophobic contributions to the Pickering effect. We have clarified this with this statement in the manuscript: *"However, hydrophobic interactions could be responsible for maintaining the stability of the Pickering effect mediated by the N-terminal domain through electrostatic interactions leading to inhibition of coalescence."*

3. Page 11, *"The surprising observation that only 19% of the total αS partitions into TDP-43PrLD – RNA droplets prompted us to investigate the partitioning behavior by confocal microscopy."*

The reviewer doesn't quite grasp why it's surprising that only 19% of αS is incorporated into the droplets. Could the authors explain in more detail?

Response: Protein concentration in the dense phase is always higher than in the dilute phase. Based on a few previous papers [3] typically the relative protein concentration in the dense phase is ~1000-fold excess than the dilute phase. However, based on our measurements, the dense phase αS is actually 5-fold lower compared to the dilute phase. We conjecture that this is due to the Pickering effect of αS and the chemistry of TDP-43PrLD – RNA interface driven by specific molecular determinants. As described in the response for question (2) above, to clarify this point further, the following statement has been incorporated in the manuscript: *"The partitioning of αS not showing significant quantitative change despite varying stoichiometry could be attributed to the protein's preferential localization on the surface of preformed TDP-43PrLD droplets preventing further recruitment from the bulk/dilute phase."*

4. Page 11, *"This was also apparent in the enhanced images and their intensity plots that continued to show Pickering effects (Fig. 4h-j)." From as far as I see Fig.4h-j, it appears that αS has penetrated considerably into the interior of the droplets. I'm doubtful if this can still be said to be a result of the Pickering.*

Response: We understand the reviewer's concern. We recognize that we did not adequately describe the possible penetration of αS after 24 h of incubation. First, although the coalescence of the droplets is significantly attenuated by the addition of αS (Fig 4g), it is not eliminated after 24 h. Some droplets show coalescence (Fig 4h, i), which is less than 10% in our assessment. Although αS seems to be more embedded within the droplets after 24h, it does not diffuse into the droplets uniformly and completely (Fig 4h-i), which is evident from the intensity profiles. To make this point clearer we have included this statement in the Results section: *"Approximately less than 10% of the droplets showed coalescence, and initial Pickering effect of αS on the surface slowly diffused to engulf the droplets. The engulfment was especially pronounced after 24 hours but αS failed to diffuse completely within the droplet"*. We also remark on this in the Discussion section with, *"The ability of αS to act as a Pickering agent localizing along the droplet's outer surface initially which changes to partial or full engulfment of the droplets, is new and intriguing"*.

5. Page 11, *"However, the 'engulfment' of the droplets was markedly decreased with αS as compared to the monomers possibly suggesting a limitation by the aggregated state of the protein for partitioning" I'm not sure what the term 'engulfment' refers to. It should be explained in the confocal microscopy figure, perhaps with arrows indicating what is being referred to.*

Response: We have now clarified the term 'engulfment' in the result section by this modified statement: *"Approximately less than 10% of the droplets showed coalescence, and the initial Pickering effect of αS on the surface slowly diffused to engulf the droplets by uniformly partitioning on the surface."*

6. Page 13, *"These confocal studies coupled with the quantitative kinetic data suggest that electrostatic*

interactions could play a role in Pickering and inhibition of coalescence to a greater extent than hydrophobic interactions.” As the authors have mentioned, the droplets coalesced 24 hours later. Therefore, the conclusion that electrostatic interactions contribute to the ‘inhibition of coalescence’ sounds questionable.

Response: As described in the previous points, we acknowledge that the coalescence of the droplets was not inhibited fully and some droplet coalescence was observed. However, it is clear from the data that the electrostatic interactions are involved in the Pickering and inhibition of coalescence more than the hydrophobic effects of α S. This has been clarified with the statement, “*These confocal studies coupled with the quantitative kinetic data suggest that electrostatic interactions could play a role in the Pickering effects and inhibition of coalescence to a greater extent than hydrophobic interactions, which also seem to contribute to some extent.*”

7. Page 15, “The sample containing α S showed an increase in ThT from 20 hours (*j*; Fig. 6c) as opposed to the sample without α S which did not show an increase in ThT intensity even after 50 hours (*j*; Fig. 6c). Incubation of the sedimented droplets in the presence of 5 μ M TDP-43PrLD monomers further exacerbated the lag time difference between the droplets with and without α S. The samples with α S showed a lag time of 8-9 hours (*l*; Fig. 6d) while the ones in its absence showed a lag time of ~25 hours (*l*; Fig. 6d).” Why is ThT added to the sedimented droplets? Wouldn't it be better to observe ThT at the same time as droplet formation? I think fibrils may already be forming inside the droplets, so why can't ThT fluorescence be observed until 20 hours have passed? Could it be the time it takes for ThT to permeate the droplets? Also, how does the increase in the ThT S-curve with added monomer relate to the droplets? Are fibrils forming from monomers on the droplet surface? If α S is added to the solution without droplets, wouldn't such an increase in ThT fluorescence be observed?

Response: We thank the reviewer for pointing out several important questions. We would like to clarify these. (i) First, it is clear from the data that α S monomers are at higher concentrations outside the droplets, and therefore, we surmised that the addition of ThT directly into the reaction might report on the fibril formation either by TDP-43/ α S heterotypic or α S homotypic aggregation in the bulk phase confounding the inference [1]. Therefore, we decided to add ThT directly into the isolated droplets instead of the reaction containing both phases. (ii) In part, we agree with the reviewer regarding the permeation of ThT. We monitored ThT fluorescence up to 50 h and observed that the fluorescence emerges after ~30h (Fig 6c). The observed lag time could have a contribution from the time taken for ThT to diffuse into the droplets along with aggregation occurring on the surface. containing TDP-43PrLD and RNA with and without α S were centrifuged at 0 h of reaction and the aggregation inside the droplets was monitored until 50 h using ThT fluorescence. We have now included this possibility in the text with, “*The observed lag time could be due to the combined effects of α S-seeded TDP-43PrLD aggregation as well as the time taken to ThT to permeate into the droplets.*” (iii) We would like to clarify that the addition of monomers to the droplets was to exacerbate the observations from samples with the droplets alone and to confirm that the presence of α S seeds nucleate TDP-43 aggregation. Clearly, TDP-43PrLD droplets in the absence of α S (black, Fig 6c, right) seeded TDP-43PrLD aggregation but at a significantly lower rate than the ones with α S confirming our conclusions. Therefore, the purpose of this experiment is to understand the role of α S in the TDP-43PrLD – RNA droplets in cross-seeding TDP-43PrLD monomers. To answer the second question, the addition of α S to the solution without droplets also leads to an increase in ThT fluorescence but with a lag time over weeks as mentioned previously in ours and other work [1].

8. Page 15, “The presence of RNA in the heterotypic fibrils emerging from the droplets was also confirmed by ethidium bromide staining (Fig S5b).” How can the fibrils be determined to be heterotypic? Evidence should be provided.

Response: This is an important question and requires some clarification of nomenclature. The term “heterotypic fibrils” is used for amyloids comprised of two or more proteins irrespective of their

stoichiometry. For example, the term “heterotypic” is used for amyloid fibrils cross-seeded with one protein seed to the monomer of another in which case, the fibril seed of one protein acts as a template to enhance the aggregation of a second protein, which can result in the formation of aggregates that are primarily composed of the second protein. Yet, these are not homotypic aggregates as they contain a small proportion of the seed. Heterotypic fibrils can also be formed by two aggregating proteins in stoichiometric proportions that we have termed ‘hybrid fibrils’ [2]. This nomenclature is also followed in many papers in the field of amyloids including ours.

9. Page 17, “...as evidenced in the relatively high random coil component (1650 cm^{-1} and 1666 cm^{-1}), and low antiparallel β -sheet bands (1686 cm^{-1}) (Fig. 7b).” The reviewer cannot recognize relationship between the wavenumber and IR spectral features claimed by the author on Fig. 7b, and even Fi. 7i. The features should be indicated by arrows or other means in the figure.

Response: We have added arrows to increase the clarity of presentation and better indicate the spectral features in question.

10. Page 17, “While significant variations in fibrillar morphology were not observed (Fig. 7g), the fibril spectra exhibited marked differences, and were somewhat similar to pure α S (Fig. 7h and red; 7i).” Figs. 7e and g appear to have very different degrees of agglomeration. This makes it a bit of a stretch to claim that there is no significant difference.

Response: We agree with the reviewer that the extent of the fibrillar network observed is different across the samples. However, we did not find any specific morphological differences among fibrils, such as twisted and flat fibrils, as typically observed in electron microscopy. We have not attempted to discern between the samples based on the extent of fibrillar network formation. We have clarified this in the revised manuscript: “The degree of fibrillar network formation was found to vary across the samples. However, we did not observe morphological variations between fibrillar aggregates, such as twisted and flat morphologies, as typically observed in electron microscopy [4]. We have not attempted to discern between the samples based on the extent of fibrillar network formation.”

11. Since Figure 7g shows a large aggregation of fibrils, the AFM probe should not detect the vibration from only a single fibril directly under the probe. So, if some fibrils were overlapped, wouldn't the IR spectrum be a mixture of both effects? Thus the reviewer considers what can be said about secondary structure from this experiment is rather uncertain. If the author could map the IR spectrum peaks at specific wavelengths, they would be able to show how much the structure differs from fibril to fibril and the inhomogeneity of fibrils. It might also be possible to show differences in secondary structure by showing a slightly more enlarged, higher resolution AFM image to show differences in fibril morphology.

Response: We agree with the reviewer that for samples where a dense fibrillar network is formed, it is challenging to obtain IR spectra that is representative of a single isolated fibril. The spectra in this case are better interpreted as arising from fibrillar components that are part of a network. However, AFM-IR measurements can still distinguish the morphological differences of aggregates like oligomer and fibril and unequivocally provide their specific secondary structure, whereas a bulk measurement like FTIR would exhibit statistically averaged out structural data combining oligomers and fibrils together. Furthermore, it can provide structural insights within different locations of a fibrillar network. Thus, the spectra reported herein are reflective of the secondary structure specifically the fibrillar phase, which we compare across different samples. This approach is valid under the assumption that the spectra from different spatial locations of a fibrillar network, and as an extension, of different components constituting that network, are not significantly different. We have verified this by acquiring spatially resolved spectra from multiple locations, as shown in Figure S6. To avoid any biases in our spectral interpretation, we have used MCR-ALS, which deconvolutes the spectra without any *a-priori* input of component spectra. The output of the analysis where the weights of four component spectra representing four structural motifs have been shown in Fig. 7i-k. The differences in weights directly indicate the variety of secondary structural distribution among samples.

We thank the reviewer for pointing out the possibility of acquiring spatial maps at specific IR frequencies to assess the heterogeneity of fibrillar networks. However, in our opinion, there are challenges that do not always allow for facile spatial mapping/imaging of secondary structures over a fibrillar network. Firstly, IR spectra of proteins arise from overlaps of multiple secondary structural components, and intensities at single wavelengths/frequencies cannot be deconvoluted to determine the exact contribution of a particular secondary structure component. Secondly, Additionally, it has been demonstrated that AFM-IR maps can also be affected by fluctuations in tip-sample interactions [5], leading to the loss of fidelity in the spectral intensity. We have therefore focused on spatially resolved spectral measurements and not images.

While a high-resolution AFM image of a larger area can potentially reveal variations in fibrillar morphologies, they would not offer any insights into the secondary structure, unless it is known a priori that different morphologies are structurally distinct. Additionally, the AFM probe that is used in our experiment has a tip radius of ~ 30 nm due to its gold coating that is necessary for measuring IR spectra. As a result, it puts a limitation on the morphological resolution attainable from these measurements. We aim to address this limitation in future work by acquiring additional high-resolution AFM images and acquiring AFM-IR spectra of the same spatial locations through image registration. Unfortunately, these measurements are beyond the scope of this work. We have clarified the above points in the revised manuscript, as follows: *“It should be noted that in the presence of fibrillar networks, such as those observed here, the AFM-IR spectral measurements are limited by the sample morphology, and are reflective of fibrillar components of the network, and not isolated, individual fibrils. Nonetheless, this is still a significant improvement over conventional spatially averaged techniques like FTIR, since we can unequivocally attribute the spectral features to fibrillar morphologies and not oligomers or other non-fibrillar aggregates. This approach is valid under the assumption that the spectra from different spatial locations of a fibrillar network, and as an extension, of different components constituting that network, are not significantly different. We have verified this by acquiring spatially resolved spectra from multiple locations, as shown in Figure S6. Recent developments in mitigating the effects of sample mechanical*

properties on the AFM-IR (*Anal. Chem.* 2018, 90, 15, 8845–8855, AND <https://doi.org/10.1073/pnas.2210516119>) can enable the acquisition of hyperspectral spatial maps of fibrillar aggregates, which can potentially reveal additional insights into structural variations within networks. Combining AFM-IR with additional high-resolution AFM can also identify subtle morphological variations within a network and their correlation, if any, with secondary structure. We hope to pursue such experimental strategies in future work.”

Minor points

1. Page 4, “Our results, obtained from a combination of high-resolution microscopy and “high-resolution microscopy” should be “Confocal optical microscopy”. Moreover, for ‘nanoscale, spatially-resolved AFM-IR spectroscopy’, the term ‘spatially-resolved’ in particular feels out of place since it’s not doing mapping. It should simply be “AFM-IR spectroscopy’.”

Response: We have made the suggested changes in the revised manuscript. However, we would like to point out that even though we are not acquiring IR images with AFM-IR, the spectra are spatially resolved, since we are measuring spectra at specific locations on a fibrillar network, and not of the entire network/sample.

2. Page 6, “...gelation (■; Fig. 1c). In contrast, α S showed no recovery in these puncta which suggests that α S could be in a more gelated, aggregated state (■; Fig. 1c).” “Fig. 1c” should be “Fig. 1b”.

Response: Fixed.

3. Page 8, “... μ g/mL in the absence of α S (●; Fig. 3g).” Color of the solid circles in Fig. 3 is inconsistent between the main text, the figure captions, and the graph.

Response: Fixed.

4. Page 8, “...above and below UCST (4 and 37 °C) showed no changes..) “4” should be “25”.

Response: Fixed.

5. Page 13, “To determine what sequence properties in α S are responsible for the observed Pickering and coarsening effects of TDP-43PrLD – RNA droplets, truncation constructs α S Δ NTD and α S Δ NAC were designed. To broadly identify the effects of two important domains of α S; the N-terminal domain (NTD; 1-60) is highly charged and the central amyloid region (NAC; 61-95) is rich in hydrophobic residues and therefore, deletion of these regions could provide clues for their contributions in Pickering. The data for the deletion mutant is already presented in Fig. 3i. Therefore, this explanation should be given before referring to Fig 3i.

Response: We understand the reviewer’s concern regarding the appropriate placement of the aforementioned statement in the text. Therefore, we deleted this statement from page 13 and inserted in the main text before Fig 3i.

6. Fig 7 a, d, e, g. The Z color bar should be marked with a unit.

Response: Fixed.

Reviewer #3:

This is an interesting and important study dedicated to the in-depth analysis of the peculiarities of the interactions between TDP-43PrLD and alpha-synuclein using cell culture models and in vitro TDP-43PrLD-RNA liquid droplets. The authors showed that the alpha-synuclein forms clusters on the surface of TDP-43PrLD – RNA droplets and emulsifies them. They also showed that such alpha-synuclein-driven

emulsification nucleates formation of the heterotypic amyloids. Such heterotypic amyloids are characterized by specific structural features, which are different from those of fibrils derived from the homogeneous solutions. The authors hypothesized that in this system, alpha-synuclein serves as a Pickering agent leading to the formation of a Ramsden or Pickering emulsion; i.e., emulsion stabilized by solid particles locating to the droplet surface. This work adds significantly to the field and will have a noticeable impact. Although the manuscript is generally well-written and concise, it has some linguistic issues and requires careful editing and proofreading.

Response: Thank you for your feedback and finding our work interesting. We have edited the manuscript for clarity and brevity.

1. Dhakal, S., et al., *Prion-like C-terminal domain of TDP-43 and α -Synuclein interact synergistically to generate neurotoxic hybrid fibrils*. J. Mol. Biol., 2021. **433**(10): p. 166953.
2. Dhakal, S., et al., *Distinct neurotoxic TDP-43 fibril polymorphs are generated by heterotypic interactions with α -Synuclein*. J Biol Chem., 2022. **298**(11).
3. Yokosawa, K., et al., *Quantification of the concentration in a droplet formed by liquid–liquid phase separation of G-quadruplex-forming RNA*. Chemical Physics Letters, 2023: p. 140634.
4. Tycko, R., *Amyloid polymorphism: structural basis and neurobiological relevance*. Neuron, 2015. **86**(3): p. 632-645.
5. Kenkel, S., et al., *Probe–sample interaction-independent atomic force microscopy–infrared spectroscopy: toward robust nanoscale compositional mapping*. Analytical chemistry, 2018. **90**(15): p. 8845-8855.

REVIEWERS' COMMENTS:

Reviewer #1 (Remarks to the Author):

The authors have addressed all my concerns and the manuscript can now be accepted for publication.

Reviewer #2 (Remarks to the Author):

The authors have diligently and effectively addressed the queries raised by the reviewers, leading to a substantial enhancement of the manuscript. I express my satisfaction with the revisions and responses provided, concluding that the manuscript now meets the requisite standards for publication.